# On the Explicit Role of Initialization on the Convergence and Generalization Properties of Overparametrized Linear Networks

## Abstract

Neural networks trained via gradient descent with random initialization and without any regularization enjoy good generalization performance in practice despite being highly overparametrized. A promising direction to explain this phenomenon is the *Neural Tangent Kernel* (NTK), which characterizes the implicit regularization effect of gradient flow/descent on infinitely wide neural networks with random initialization. However, a non-asymptotic analysis that connects generalization performance, initialization, and optimization for finite width networks remains elusive. In this paper, we present a novel analysis of overparametrized single-hidden layer linear networks, which formally connects initialization, optimization, and overparametrization with generalization performance. We exploit the fact that gradient flow preserves a certain matrix that characterizes the *imbalance* of the network weights, to show that the squared loss converges exponentially at a rate that depends on the level of imbalance of the initialization. Such guarantees on the convergence rate allow us to show that large hidden layer width, together with (properly scaled) random initialization, implicitly constrains the dynamics of the network parameters to be close to a low-dimensional manifold. In turn, minimizing the loss over this manifold leads to solutions with good generalization, which correspond to the min-norm solution in the linear case. Finally, we derive a novel $\mathcal{O}(h^{-1/2})$ upper-bound on the operator norm distance between the trained network and the min-norm solution, where $h$ is the hidden layer width.

## 1 Introduction

Neural networks have shown excellent empirical performance in many application domains such as vision (Krizhevsky et al., 2012; Rawat & Wang, 2017), speech (Hinton et al., 2012; Graves et al., 2013) and video games (Silver et al., 2016; Vinyals et al., 2017). Among the many unexplained mysteries behind this success is the fact that gradient descent with random initialization and without explicit regularization enjoys good generalization performance despite being highly overparametrized.

A promising attempt to explain this phenomena is the *Neural Tangent Kernel* (NTK) (Jacot et al., 2018), which characterizes the implicit regularization effect of gradient flow/descent on infinitely wide neural networks with random initialization. Precisely, under this infinite width assumption, a proper initialization, together with gradient flow training, can be understood as a kernel gradient flow (NTK flow) of a functional that is constrained on a manifold that guarantees good generalization performance. The analysis further admits extensions to the finite width (Arora et al., 2019b; Buchanan et al., 2020). The core of the argument, illustrated in Figure 1, amounts to showing that (properly scaled) random initialization of networks with sufficiently large width, leads to trajectories that are, in some sense initialized close to the aforementioned manifold. Thus, approximately good initialization, together with acceleration, ensures that the training stays close to the NTK Flow.

Such analysis, however, leads to bounds on the network width that significantly exceed practical values (Buchanan et al., 2020), and seems to suggest the need of acceleration to achieve generalization. This motivates the following questions:

- *Is the kernel regime, which requires impractical bounds on the network width, necessary to achieve good generalization?*

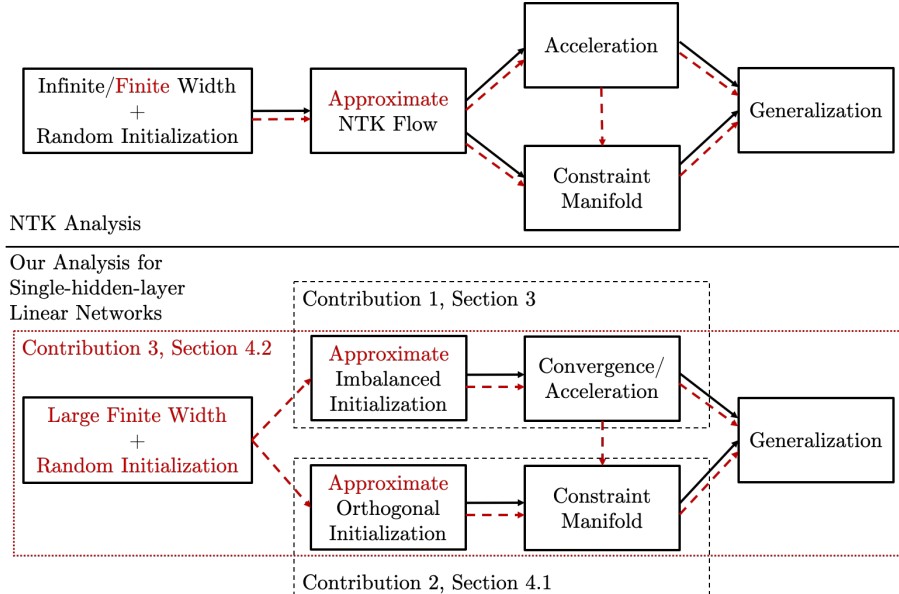

Figure 1: Comparing our analysis to asymptotic/non-asymptotic NTK analysis.

- *Does generalization depends explicitly on acceleration? Or is acceleration required only due to the choosing an initialization outside the good generalization manifold?*

For the simplified, yet certainly non-trivial, single-hidden layer linear network setting, this paper finds an answer to these questions.

**Contributions.** We present a novel analysis of the gradient flow dynamics of overparametrized single-hidden layer linear networks, which provides disentangled conditions on initialization that lead to acceleration and generalization. Specifically, we show that imbalanced initialization ensures acceleration, while orthogonal initialization ensures that trajectories remain close to the generalization manifold. Interestingly, properly scaled random initialization of moderately wide networks is sufficient to ensure that initialization is approximately imbalanced and orthogonal, yet it is not necessary for either. More specifically, as illustrated in Figure 1, this paper makes the following contributions:

1. We show first that gradient flow on the squared-$l_2$ loss preserves a certain matrix-valued quantity, akin to constants of motion in mechanics or conservation laws of physics, that measures the *imbalance* of the network weights. Notably, some level of imbalance, measured by certain singular value of the imbalance matrix and defined at initialization, is sufficient to guarantee the exponential rate of convergence of the loss. Our analysis is non-probabilistic and valid under very mild assumptions, satisfied by moderately wide single-hidden-layer linear networks.

2. We characterize the existence of a low-dimensional manifold defined by a specific orthogonality condition on the parameters, which is invariant under the gradient flow. All trajectories within this manifold lead to a unique (w.r.t the end-to-end function) minimizer with good generalization performance, which corresponds to the min-norm solution. As a result, initializing the network within this manifold guarantees good generalization.

3. We further show that by randomly initializing the network weights using $\mathcal{N}(0, 1/h)$ (where $h$ is the hidden layer width), an initialization setting related to the kernel regime (see Appendix E), one can approximately satisfy both our sufficient imbalance and orthogonality conditions with high probability. Notably, the inaccurate initialization relative to the good generalization manifold requires acceleration to control the generalization error. In the context of NTK, our result further provide for linear networks a novel $\mathcal{O}\left(h^{-1/2}\right)$ upper-bound on the operator norm distance between the trained network and the min-norm solution. To the best of our knowledge, this is the

first non-asymptotic bound regarding the generalization property of wide linear networks under random initialization in the global sense.

**Notation.** For a matrix $A$, we let $A^T$ denote its transpose, $\mathrm{tr}(A)$ denote its trace, $\lambda_i(A)$ and $\sigma_i(A)$ denote its $i$-th eigenvalue and $i$-th singular value, respectively, in decreasing order (when adequate). We let $[A]_{ij}$, $[A]_{i,:}$, and $[A]_{:,j}$ denote the $(i,j)$-th element, the $i$-th row and the $j$-th column of $A$, respectively. We also let $\|A\|_2$ and $\|A\|_F$ denote the spectral norm and the Frobenius norm of $A$, respectively. For a scalar-valued or matrix-valued function of time, $F(t)$, we let $\dot{F} = \dot{F}(t) = \frac{d}{dt}F(t)$ denote its time derivative. Additionally, we let $I_n$ denote the identity matrix of order $n$ and $\mathcal{N}(\mu, \sigma^2)$ denote the normal distribution with mean $\mu$ and variance $\sigma^2$.

## 2 RELATED WORK

**Wide neural networks**. There is a rich line of research that studies the convergence (Du et al., 2019b;a; Du & Hu, 2019; Allen-Zhu et al., 2019b) and generalization (Allen-Zhu et al., 2019a; Arora et al., 2019a;b; Li & Liang, 2018; Cao & Gu, 2019; Buchanan et al., 2020) of wide neural networks with random initialization. The behavior of such networks in their infinite width limit can be characterized by the *Neural Tangent Kernel* (NTK) (Jacot et al., 2018). With the concept of NTK, heuristically, training wide neural networks can be approximately viewed as kernel regression under gradient flow/descent (Arora et al., 2019b), hence the convergence and generalization can be understood by studying the non-asymptotic results regarding the equivalence of finite width networks to their infinite limit (Du et al., 2019b;a; Du & Hu, 2019; Allen-Zhu et al., 2019b; Arora et al., 2019a;b; Buchanan et al., 2020). More generally, such non-asymptotic results are related to the "lazy training" (Chizat et al., 2019; Du et al., 2019a; Allen-Zhu et al., 2019b), where the network weights do not deviate too much from its initialization during training. Our results for wide linear networks presented in Section 4.2 do not follow the NTK analysis, but provide an alternative, presumably more general view on the effect of random initialization when the hidden layer is sufficiently wide.

**Convergence of linear networks**. Convergence in overparametrized linear networks has been studied for both gradient flow (Saxe et al., 2014) and gradient descent (Bartlett et al., 2018; Arora et al., 2018a;b). In the kernel regime, Du & Hu (2019) applied the analysis of convergence of wide neural networks (Du et al., 2019b) to deep linear networks. Aside from the large hidden layer width and random initialization assumptions, Saxe et al. (2014) analyzed the trajectory of network parameters under spectral initialization, while Bartlett et al. (2018) studied the case of identity initialization. Although the fact that the imbalance is conservative under gradient flow has been exploited in Arora et al. (2018b;a), they consider the case of balanced initialization only to simplify the learning dynamics, and additional conditions are required for convergence. Most works mentioned above considered specific, often data-dependent, types of initialization that makes the learning dynamics tractable. Our result, based on an imbalance measure, is data-agnostic and satisfied under a wide range of random initialization schemes, see lemmas 1 and F.2.

**Min-norm solution in high-dimension linear regression**. For high-dimensional under-determined linear regression, the asymptotic generalization error of min-norm solution has been studied in Hastie et al. (2019). In Bartlett et al. (2020); Mei & Montanari (2019), the min-norm solution was proved to have near-optimal generalization performance under mild assumptions on the data model. For our purpose, we study the generalization of a trained linear network as its distance to the min-norm solution. In this way, we refer to solutions with good generalization performance, as those with small distance to the min-norm solution.

## 3 CONVERGENCE RATE OF GRADIENT FLOW FOR SINGLE-HIDDEN-LAYER LINEAR NETWORKS

In this section we study the convergence of gradient flow on squared $l_2$-loss for single-hidden-layer linear networks. Given training data of $n$ samples $\{x^{(i)}, y^{(i)}\}_{i=1}^n$ with $x^{(i)} \in \mathbb{R}^D, y^{(i)} \in \mathbb{R}^m$, we aim to solve the linear regression problem

$$\min_{\Theta \in \mathbb{R}^{D \times m}} \mathcal{L} = \frac{1}{2} \sum_{i=1}^n (y^{(i)} - \Theta^T x^{(i)})^2 \,, \tag{1}$$

by training a single-hidden-layer linear network $y = f(x; V, U) = VU^T x, V = \mathbb{R}^{m \times h}, U \in \mathbb{R}^{D \times h}$ with gradient flow, or equivalently, gradient descent with "infinitesimal step size", here $h$ is the hidden layer width. We consider an *overparametrized* model such that $h \geq \min\{m, D\}$.

We rewrite the loss function with respect to our parameters $V, U$ as

$$\mathcal{L}(V, U) = \frac{1}{2} \sum_{i=1}^{n} (y^{(i)} - VU^T x^{(i)})^2 = \frac{1}{2} \|Y - XUV^T\|_F^2 \,, \tag{2}$$

where $Y = [y^{(1)}, \cdots, y^{(n)}]^T, X = [x^{(1)}, \cdots, x^{(n)}]^T$ are concatenations of the training data in rows. Assuming that the input data $X$ has full rank, we consider the under-determined case $D > n$ for our regression problem, i.e., there are infinitely many solutions $\Theta^*$ to achieve zero loss of (1). With minor a reformulation, our convergence result works for the case where the input data $X$ is rank deficient. We refer the reader to Appendix B.

We will show that under certain conditions, the trajectory of the loss function $\mathcal{L}(t) = \mathcal{L}(V(t), U(t))$ under gradient flow of (2), i.e.,

$$\dot{V}(t) = -\frac{\partial}{\partial V} \mathcal{L}(V(t), U(t)) \,, \dot{U}(t) = -\frac{\partial}{\partial U} \mathcal{L}(V(t), U(t)) \,, \tag{3}$$

converges to 0 exponentially, and that proper initialization of $U(0), V(0)$ controls the convergence rate via a time-invariant matrix-valued term, the *imbalance* of the network.

## 3.1 REPARAMETRIZATION OF GRADIENT FLOW

Given $D > n = \text{rank}(X)$, the singular value decomposition of $X$ is given by

$$X = W \begin{bmatrix} \Sigma_x^{1/2} & 0 \end{bmatrix} \begin{bmatrix} \Phi_1^T \\ \Phi_2^T \end{bmatrix} \,, W \in \mathbb{R}^{n \times n} \,, \Phi_1 \in \mathbb{R}^{D \times n} \,, \Phi_2 \in \mathbb{R}^{D \times (D-n)} \,, \tag{4}$$

with $WW^T = W^T W = \Phi_1^T \Phi_1 = I_n \,, \Phi_2^T \Phi_2 = I_{D-n} \,, \Phi_1^T \Phi_2 = 0$, and $\Phi_1 \Phi_1^T + \Phi_2 \Phi_2^T = I_D$.

Notice that
$$U = I_D U = (\Phi_1 \Phi_1^T + \Phi_2 \Phi_2^T) U = \Phi_1 \Phi_1^T U + \Phi_2 \Phi_2^T U \,,$$

hence we can reparametrize $U$ as $(U_1, U_2)$ using the bijection $g : \mathbb{R}^{n \times h} \times \mathbb{R}^{(D-n) \times h} \to \mathbb{R}^{D \times h} \,, U = g(U_1, U_2) = \Phi_1 U_1 + \Phi_2 U_2$, with inverse $(U_1, U_2) = g^{-1}(U) = (\Phi_1^T U, \Phi_2^T U)$.

We write the gradient flow in (3) explicitly as

$$\dot{V}(t) = \left(Y - XU(t)V^T(t)\right)^T XU(t) = E^T(t) \Sigma_x^{1/2} \Phi_1^T U(t) \,, \tag{5a}$$

$$\dot{U}(t) = X^T \left(Y - XU(t)V^T(t)\right) V(t) = \Phi_1 \Sigma_x^{1/2} E(t) V(t) \,, \tag{5b}$$

based on the SVD of data $X$ in (4), where

$$E(t) = E(V(t), U_1(t)) = W^T Y - \Sigma_x^{1/2} U_1(t) V^T(t) \,, \tag{6}$$

is defined to be the *error*. Then from (5a)(5b) we obtain the dynamics in parameter space $(V, U_1, U_2)$ as

$$\dot{V}(t) = E^T(t) \Sigma_x^{1/2} U_1(t) \,, \dot{U}_1(t) = \Sigma_x^{1/2} E(t) V(t) \,, \dot{U}_2(t) = 0 \,. \tag{7}$$

Notice that since $W$ is orthogonal, we have

$$\mathcal{L}(t) = \frac{1}{2} \|Y - XU(t)V^T(t)\|_F^2 = \frac{1}{2} \|WE(t)\|_F^2 = \frac{1}{2} \|E(t)\|_F^2 \,. \tag{8}$$

Therefore it suffices to analyze the convergence rate of the error $E(t)$ under the dynamics of $V(t), U_1(t)$ in (7). As we mentioned in Section 1, the exponential convergence of $E(t)$, or equivalently the loss function $\mathcal{L}(t)$ is crucial for our analysis for generalization, in the sense that exponential convergence ensures that the parameters do not deviate much away from the manifold of our interest, which we will discuss in Section 4, so that good properties from the initialization are approximately preserved during training.

### 3.2 IMBALANCE AND CONVERGENCE RATE OF THE ERROR

We define the *imbalance* of the single-hidden-layer linear network under input data $X$ as

$$Imbalance:\ U_1^T U_1 - V^T V\,. \tag{9}$$

The imbalance term is time-invariant under gradient flow, as stated in the following claim

**Claim.** *Under continuous dynamics* (7)*, we have* $\frac{d}{dt}[U_1^T(t)U_1(t) - V^T(t)V(t)] \equiv 0$.

*Proof.* Under (7), we compute the time derivative of $U_1^T(t)U_1(t)$ and $V^T(t)V(t)$ as

$$\frac{d}{dt}U_1^T(t)U_1(t) = \dot{U}_1^T(t)U_1(t) + U_1^T(t)\dot{U}_1(t) = V^T(t)E^T(t)\Sigma_x^{1/2}U_1(t) + U_1^T(t)\Sigma_x^{1/2}E(t)V(t),$$

$$\frac{d}{dt}V^T(t)V(t) = V^T(t)\dot{V}(t) + \dot{V}^T(t)V(t) = V^T(t)E^T(t)\Sigma_x^{1/2}U_1(t) + U_1^T(t)\Sigma_x^{1/2}E(t)V(t)\,.$$

The right-hand side of two equations is identical, hence $\frac{d}{dt}[U_1^T(t)U_1(t) - V^T(t)V(t)] \equiv 0$. □

The imbalance is a $h \times h$ matrix with rank at most $m + n$, the rank of imbalance characterizes how much the row spaces of $U_1$ and $V$ are misaligned. We show that a rank-$(m + n - 1)$ imbalance is sufficient for exponential convergence of the error $E(t)$, or equivalently, the loss function.

Now we present our result regarding convergence of the error. (see Appendix D for the proof).

**Theorem 1** (Convergence of linear networks with sufficient rank of imbalance)**.** *Suppose* $h \geq m + n - 1$. *Let* $V(t), U_1(t), t > 0$ *be the trajectory of continuous dynamics* (7) *starting from some* $V(0), U_1(0)$. *If*

$$\sigma_{n+m-1}\left(U_1^T(0)U_1(0) - V^T(0)V(0)\right) = c > 0\,, \tag{10}$$

*then for* $E(t)$ *defined in* (6)*, we have*

$$\|E(t)\|_F^2 \leq \exp\left(-2\sigma_n(\Sigma_x)ct\right)\|E(0)\|_F^2,\ \forall t > 0\,. \tag{11}$$

*Additionally,* $V(t), U_1(t), t > 0$ *converges to some equilibrium point* $(V(\infty), U_1(\infty))$ *such that* $E(V(\infty), U_1(\infty)) = 0$.

The fact that the imbalance is preserved under gradient flow has been exploited in Arora et al. (2018a;b), where imbalance is assumed to be zero (or small), such that the learning dynamics can be expressed in closed form with respect to the end-to-end matrix. This analysis, requires, however, additional assumptions on the initialization of the end-to-end matrix for acceleration. Similarly, though in a more general setting, Du et al. (2018) showed that the imbalance is preserved, and proves convergence under a small imbalance assumption. Acceleration (exponential rate), however, is not guaranteed. Exploiting imbalance for guaranteeing acceleration was first presented in Saxe et al. (2014), under a spectral initialization assumption. In contrast, Theorem 1 shows acceleration without the spectral initialization condition. Rather, we show that under very mild conditions on the alignment between the initialization and the data, acceleration is achieved.

Such good choice of initialization, at first glance, seems to largely depend on the data given the definition of the imbalance. However, we show in the next section that for sufficiently wide networks with random initialization, the imbalance has rank at least $n + m$ with high probability, for any data matrix $X$, hence exponential convergence is almost guaranteed when training such networks. Later we will illustrate how such convergence also affects the generalization of the trained network.

The dependence on $\sigma_n(\Sigma_x)$ has been also appeared in Du & Hu (2019), where a linear convergence rate of gradient descent was shown for a multi-layer linear networks. Their proof followed the same high-level procedure as in showing convergence for networks with nonlinear activation (Du et al., 2019b;a), which relied on showing that the Gram matrix is close to its initialization during training. For our result, although it is provided for single-hidden-layer linear networks, we essentially lower bound the smallest eigenvalue of the Gram matrix at any time $t$ by a fixed constant that only depends on the initialization.

We end the section by noting that our result is not restricted to the case that $X$ is full rank. In Appendix B, we show that similar result hold for the case that $X$ is rank deficient, with minor reformulations. In that case, we only require $h \geq d + m - 1$, and the singular value $\sigma_{d+m-1}(U_1^T(0)U_1(0) - V^T(0)V(0))$ replaces what in shown in (10), where $d$ is the rank of $X$. In addition, we present and discuss the numerical simulation regarding Theorem 1 in Appendix A.

## 4    GENERALIZATION OF SINGLE-HIDDEN-LAYER LINEAR NETWORK

In this section, we study the generalization properties of trained single-hidden-layer linear networks under gradient flow. Assuming that $D > n = \text{rank}(X)$, the regression problem (1) has infinitely many solutions $\Theta^*$ that achieve zero loss. Among all these solutions, one that is of particular interest in high-dimensional linear regression is the *minimum norm solution* (min-norm solution)

$$\hat{\Theta} = \underset{\Theta \in \mathbb{R}^{D \times m}}{\arg\min}\{\|\Theta\|_2 : Y - X\Theta = 0\} = X^T(XX^T)^{-1}Y, \tag{12}$$

which has near-optimal generalization error for suitable data models, as shown in (Bartlett et al., 2020; Mei & Montanari, 2019). Here, we study conditions under which our trained network is equal or close to the min-norm solution by showing how the initialization explicitly controls the trajectory of the training parameters to be exactly (or approximately) confined within some low-dimensional manifold. In turn, minimizing the loss over this manifold leads to the min-norm solution.

Moreover, our analysis on constrained learning applies to wide single-hidden-layer linear networks with random initialization, whose infinite width limit is equivalent to the kernel predictor with linear kernel $K(x, x') = x^T x'$, as suggested by Jacot et al. (2018). One can easily check that such a kernel predictor is the min-norm solution $\hat{\Theta}$. In addition, we show that the operator norm distance between trained finite width single-hidden-layer linear network and the min-norm solution is upper bounded by a $\mathcal{O}(h^{-1/2})$ term with high probability over random initialization.

### 4.1    DECOMPOSITION OF TRAINED NETWORK

To begin with, notice that the linear operator $UV^T \in \mathbb{R}^{D \times m}$ associated with the single-hidden-layer linear network can be decomposed according to the data matrix $X$ as

$$UV^T = (\Phi_1 \Phi_1^T + \Phi_2 \Phi_2^T)UV^T = \Phi_1 U_1 V^T + \Phi_2 U_2 V^T, \tag{13}$$

where $\Phi_1, \Phi_2, U_1, U_2$ are previously defined in Section 3. Here $[UV^T]_{:,j}$, i.e., the $j$-th column of $UV^T$, is the linear predictor for the $j$-th output $y_j$, and is decomposed into two components within complementary subspaces $\text{span}(\Phi_1)$ and $\text{span}(\Phi_2)$. Moreover $[U_1 V^T]_{:,j}$ is the coordinate of $[UV^T]_{:,j}$ w.r.t. the orthonormal basis as the columns of $\Phi_1$, and similarly $[U_2 V^T]_{:,j}$ is the coordinate w.r.t. basis $\Phi_2$. Clearly, under gradient flow (3), the trajectory $(U(t)V(t)^T, t > 0)$ is fully determined by the trajectory $(U_1(t)V^T(t), U_2(t)V^T(t), t > 0)$, which is governed by the dynamics (7).

**Convergence of Training Parameters.** We have derived useful results regarding $U_1(t)V^T(t)$ for $t > 0$ in Section 3. By Theorem 1, if the rank of the imbalance matrix is large enough, $U_1(t)V^T(t)$ converges to some $U_1(\infty)V^T(\infty)$ and the stationary point satisfies $W^T Y - \Sigma_x^{1/2} U_1(\infty)V^T(\infty) = 0$, which implies $U_1(\infty)V^T(\infty) = \Sigma_x^{-1/2} W^T Y$. Then it is easy to check that

$$\Phi_1 U_1(\infty)V^T(\infty) = \Phi_1 \Sigma_x^{-1/2} W^T Y = X^T(XX^T)^{-1}Y = \hat{\Theta}. \tag{14}$$

In other words, the projected trajectory (in columns) of $U(t)V^T(t)$ onto $\text{span}(\Phi_1)$ converges exactly to the min-norm solution.

For $U_2(t)V^T(t)$, notice that $\dot{U}_2(t) = 0$ in dynamics (7), hence $U_2(t) = U_2(0), \forall t > 0$. Then under sufficient rank of imbalance, $U(t)V^T(t)$ converges to some $U(\infty)V^T(\infty)$ and

$$U(\infty)V^T(\infty) = \Phi_1 U_1(\infty)V^T(\infty) + \Phi_2 U_2(0)V^T(\infty) = \hat{\Theta} + \Phi_2 U_2(0)V^T(\infty).$$

Therefore $U_2(0)V^T(\infty)$ quantifies how much the trained network $U(\infty)V^T(\infty)$ is deviated from the min-norm solution $\hat{\Theta}$, and since $\Phi_2^T \Phi_2 = I_{D-n}$, we have

$$\|U(\infty)V^T(\infty) - \hat{\Theta}\|_2 = \|\Phi_2 U_2(0)V^T(\infty)\|_2 = \|U_2(0)V^T(\infty)\|_2. \tag{15}$$

**Constrained Training via Initialization.** Based on our analysis above, initializing $U_2(0)$ such that $U_2(0)V^T(\infty) = 0$ in the limit, guarantees convergence to the min-norm solution via (15). However, this is not easily achievable, as one needs to know a priori $V(\infty)$. Instead, we can show that by choosing a proper initialization, one can constrain the trajectory of the matrix $U(t)V^T(t)$

to lie identically in the set $\Phi_2^T U(t) V^T(t) \equiv 0$ for all $t \geq 0$, which is equivalent to saying that the columns of $U(t) V^T(t)$ lie in $\mathrm{span}(\Phi_1)$. Indeed, using the fact that for all $t \geq 0$, $U_2(t) = U_2(0)$ and $\Phi_2^T U(t) V^T(t) \equiv 0$ we obtain

$$0 = \Phi_2^T U(t) V^T(t) = \Phi_2^T U(\infty) V^T(\infty) = U_2(\infty) V^T(\infty) = U_2(0) V^T(\infty). \tag{16}$$

Therefore, using (15), it follows that $U(\infty) V^T(\infty) = \hat{\Theta}$, as desired.

Here, the constraint that all columns of $U(t) V^T(t)$ lie in $\mathrm{span}(\Phi_1)$ is equivalent to the constraint that $(V, U)$ is within some low-dimensional manifold in the parameter space. More importantly, such constraint on $UV^T$ is relevant to generalization: When a column of $UV^T$ is in $\mathrm{span}(\Phi_2)$, predictions are made based on features in $\mathrm{span}(\Phi_2)$. However, those features are not present in the data $X$ that spans $\Phi_1$, thus, intuitively, hurting the generalization performance.

To enforce the constraint $\Phi_2^T U(t) V^T(t) \equiv 0$, consider the dynamics of $U_2(0) V^T(t)$, or equivalently $V(t) U_2^T(0)$. From (7) we have

$$\frac{d}{dt} \begin{bmatrix} V(t) U_2^T(0) \\ U_1(t) U_2^T(0) \end{bmatrix} = \begin{bmatrix} 0 & E^T(t) \Sigma_x^{1/2} \\ \Sigma_x^{1/2} E(t) & 0 \end{bmatrix} \begin{bmatrix} V(t) U_2^T(0) \\ U_1(t) U_2^T(0) \end{bmatrix}. \tag{17}$$

The most straightforward way to enforce $V(t) U_2^T(0) = 0, \forall t > 0$ is to properly initialize $V(0), U(0)$ such that $V(0) U_2^T(0) = 0$ and $U_1(0) U_2^T(0) = 0$. To have such a proper initialization, one can

- Initialize columns of $U(0)$ in $\mathrm{span}(\Phi_1)$, which leads to $U_2(0) = 0$;
- Initialize $U(0)$ and $V(0)$ to enforce the orthogonality condition on the rows, i.e. $V(0) U^T(0) = 0$ and $U(0) U^T(0) = I_D$.

Such initialization guarantees that gradient flow is constrained within some low-dimensional manifold in the parameter space, such that any global minimizer of the loss in that manifold corresponds to the min-norm solution. Therefore, whenever the network parameters in this manifold converge, and $E(\infty) = 0$, then the solution must be the minimum-norm one. While in practice we can make the initialization exactly as above, such choice is data-dependent and requires the SVD of the data matrix $X$. Moreover, we note that while the zero initialization works for the standard linear regression case, such initialization $V(0) = 0, U(0) = 0$ is bad in the overparametrized case because it is a saddle point of the gradient flow, even though it satisfies the orthogonal condition $V(0) U_2^T(0) = 0$ and $U_1(0) U_2^T(0) = 0$.

In the next section, we show that under random initialization and sufficiently large hidden layer width $h$, these conditions on initialization are approximately satisfied, i.e., with high probability the rank of imbalance is $m + n$ and $\|V(0) U_2^T(0)\|_F, \|U_1(0) U_2^T(0)\|_F \sim \mathcal{O}(h^{-1/2})$. So that the trajectory $U(t) V^T(t), t > 0$ will be approximately constrained in the subspace as we mentioned above.

## 4.2 Wide single-hidden-layer Linear Network

We will now discuss the generalization of wide single-hidden-layer linear network with random initialization. In particular, we will show how the previously mentioned conditions for convergence and good generalization, i.e., high imbalance and orthogonality, are approximately satisfied with high probability under the following initialization

$$[U(0)]_{ij} \sim \mathcal{N}\left(0, \frac{1}{h}\right), \ 1 \leq i \leq D, 1 \leq j \leq h,$$

$$[V(0)]_{ij} \sim \mathcal{N}\left(0, \frac{1}{h}\right), \ 1 \leq i \leq m, 1 \leq j \leq h,$$

where all the entries are independent. Our analysis indeed highlights the need, within this regime, of exponential convergence to ensure good generalization.

**Remark 1.** *Our analysis can be extended to the more general case where all entries of $U(0), V(0)$ are sampled from $\mathcal{N}(0, h^{-2\alpha})$ with $\frac{1}{4} < \alpha \leq \frac{1}{2}$. For the simplicity of the presentation, we consider the particular case $\alpha = \frac{1}{2}$ in this section. Please see Appendix F for the more general result.*

Previous works Jacot et al. (2018) have suggested in the limit $h \to \infty$, the trained network is equivalent to the kernel predictor with NTK. For linear networks, the NTK is the linear kernel $K(x, x') = x^T x'$ whose corresponding kernel predictor is the min-norm solution. Therefore, as $h \to \infty$, we should expect the trained network to converge to the min-norm solution, given proper scaling of the network (Arora et al., 2019b). Combining what we have discussed regarding the convergence and generalization of linear networks with basic random matrix theory, we are able to derive the non-asymptotic $\mathcal{O}(h^{-1/2})$ bound on the operator norm distance between a trained $h$-width network under random initialization and the min-norm solution.

**Remark 2.** *We note that, both our parametrization and initialization, are at first sight different that the one used in previous works (Jacot et al., 2018; Du & Hu, 2019; Arora et al., 2019b) on NTK analysis. However, one can relate our model assumptions to the NTK ones by a rescaling of the parameters and time. In the context of this comparison, we see that our setting achieves the same as it achieves the same limiting end-to-end function, but with a rate of convergence $h$ times faster (due to the timescale rescaling). Further, our result does not rely on studying the tangent kernel of the network, hence there is significant difference between our approach to the NTK one.*

Recall in the last section, one can obtain exactly min-norm solution via proper initialization of the single-hidden-layer network. In particular, it requires 1) convergence of the error $E(t)$ to zero; and 2) the orthogonality conditions $V(0)U_2^T(0) = 0$ and $U_1(0)U_2^T(0) = 0$. Under random initialization and sufficiently large hidden layer width $h$, these two conditions are approximately satisfied. Using basic random matrix theory, one can show that with high probability, the rank of imbalance is $m + n$, which leads to (exponential) convergence of $E(t)$, and $\|V(0)U_2^T(0)\|_F, \|U_1(0)U_2^T(0)\|_F \sim \mathcal{O}(h^{-1/2})$, as stated in the following lemma (See Appendix F for the proof)

**Lemma 1.** *Given data matrix $X$. $\forall \delta \in (0, 1), \forall h > h_0 = poly\left(m, D, \frac{1}{\delta}\right)$, with probability at least $1 - \delta$ over random initializations with $[U(0)]_{ij}, [V(0)]_{ij} \sim \mathcal{N}(0, h^{-1})$, we have all the following hold.*

1. *(Sufficient rank of imbalance)*

$$\sigma_{n+m}\left(U_1^T(0)U_1(0) - V^T(0)V(0)\right) > 1 - 2\frac{\sqrt{m + D} + \frac{1}{2}\log\frac{2}{\delta}}{\sqrt{h}},\tag{18}$$

2. *(Approximate orthogonality condition)*

$$\left\|\begin{bmatrix} V(0)U_2^T(0) \\ U_1(0)U_2^T(0) \end{bmatrix}\right\|_F \le 2\sqrt{m + n}\frac{\sqrt{m + D} + \frac{1}{2}\log\frac{2}{\delta}}{\sqrt{h}},\tag{19}$$

$$\left\|U_1(0)V^T(0)\right\|_F \le 2\sqrt{m}\frac{\sqrt{m + D} + \frac{1}{2}\log\frac{2}{\delta}}{\sqrt{h}}.\tag{20}$$

Clearly, $\|V(0)U_2^T(0)\| \sim \mathcal{O}(h^{-1/2})$ alone is insufficient to have the final bound on the trained network. However, from (17) we show that as long as the error $E(t)$ converges to 0 exponentially, which is guaranteed by sufficient rank of imbalance with high probability, the final deviation from min-norm solution $\|V(\infty)U_2^T(0)\|_F$ can not exceed $C(\|V(0)U_2^T(0)\|_F + \|U_1(0)U_2^T(0)\|_F)$ for some constant $C$ that depends on the data and the convergence rate of $E(t)$, leading to the desired bound. The formal statement is summarized in the following theorem.

**Theorem 2** (Generalization of wide single-hidden-layer linear network). *Let $(V(t), U(t), t > 0)$ be a trajectory of continuous dynamics (7). Then, $\exists C > 0$, such that $\forall \delta \in (0, 1), \forall h > h_0 = poly\left(m, D, \frac{1}{\delta}, \frac{\sigma_1(\Sigma_x)}{\sigma_n^3(\Sigma_x)}\right)$, with probability $1 - \delta$ over random initializations with $[U(0)]_{ij}, [V(0)]_{ij} \sim \mathcal{N}(0, h^{-1})$, we have*

$$\|U(\infty)V^T(\infty) - \hat{\Theta}\|_2 \le 2C\sqrt{m + n}\frac{\sqrt{m + D} + \frac{1}{2}\log\frac{2}{\delta}}{\sqrt{h}},\tag{21}$$

*where $C$ depends on the data $X, Y$.*

The proof is shown in Appendix F. This is, to our best knowledge, the first non-asymptotic bound in the global (operator) sense of gradient flow trained wide neural networks under random initialization.

Although we understand that this can not be directly compared to previous works (Arora et al., 2019b; Buchanan et al., 2020) that show non-asymptotic results connecting a trained network to the kernel predictor by the NTK, using more general network structure and activation functions than that of the linear network, we believe this theorem is a clear illustration of how overparametrization, in particular the hidden layer width, together with random initialization affects the convergence and generalization, beyond the kernel regime.

To be specific, regarding the non-asymptotic analysis for wide neural networks, the concept of constrained learning presented in this section and used to prove Theorem 2 is more general than previous works (Arora et al., 2019b), where one requires sufficiently large hidden layer width such that the trajectory of the network in the function space approximately matches the trajectory in the infinite width limit. Loosely speaking, such a large width $h$ enforces the trajectory to be approximately confined within a one-dimension manifold, parametrized only by $t$, and independent of the initialization. We have shown here, however, that even for relative smaller width $h$, there is a larger dimensional manifold that provides good generalization performance. This shows that, while the kernel regime maybe sufficient, it is certainly, at least for the linear single-hidden layer network, not necessary to guarantee good generalization.

To verify Theorem 2, we present the numerical simulation regarding the implicit regularization of gradient descent on wide linear network in Appendix A. The simulation result shows that $\|U(\infty)V^T(\infty) - \hat{\Theta}\|_2$ approximately has the order $\mathcal{O}(h^{-1})$ as $h$ grows, suggesting that our non-asymptotic bound is tight in the order w.r.t. $h$. We refer interested readers to the appendix for the details of the simulation settings.

## 5 CONCLUSION

In this paper, we study the explicit role of initialization on controlling the convergence and generalization of single-hidden-layer linear networks trained under gradient flow. First of all, initializing the imbalance to have sufficient rank leads to the exponential convergence of the loss. Then proper initialization enforces the trajectory of network parameters to be exactly (or approximately) constrained in a low-dimensional manifold, over which minimizing the loss yields the min-norm solution. Combining those results, we obtain $\mathcal{O}(h^{-1/2})$ non-asymptotic bound regarding the equivalence of trained wide linear networks under random initialization to the min-norm solution. Our analysis, although on a simpler overparametrized model, formally connects overparametrization, initialization, and optimization with generalization performance. We think it is promising to translate some of the concepts such as the imbalance, and the constrained learning concept to multi-layer linear networks, and eventually to neural networks with nonlinear activations.

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

## A  NUMERICAL VERIFICATION

The scale of the linear regression we consider in the numerical section is $D = 400$, $n = 100$, and $m = 1$.

### A.1  CONVERGENCE OF SINGLE-HIDDEN-LAYER LINEAR NETWORK

**Generating training data** The synthetic training data is generated as following:

1) For data matrix $X$, first we generate $X_0 \in \mathbb{R}^{n \times D}$ with all the entries sampled from $\mathcal{N}(0, 1)$, and take its SVD $X_0 = W\Sigma^{1/2}\Phi_1$. Then we let $X = W\Phi_1$, hence we have all the singular values of $X$ being 1.

2) For $Y$, we first sample $\Theta \sim \mathcal{N}(0, D^{-1}I_D)$, and $\epsilon \sim \mathcal{N}(0, 0.01^2 I_n)$, then we let $Y = X\Theta + \epsilon$.

**Initialization and Training** We set the hidden layer width $h = 500$. We initialize $U(0), V(0)$ with $[U(0)]_{ij} \sim \mathcal{N}(0, \sigma_U^2)$, $[V(0)]_{ij} \sim \mathcal{N}(0, \sigma_V^2)$, and we consider two cases of such initialization: 1) $\sigma_U = 0.1$, $\sigma_V = 0.1$; 2) $\sigma_U = 0.5$, $\sigma_V = 0.02$. For these two cases, we run gradient descent on the averaged loss $\tilde{L} = \frac{1}{n}\|Y - XUV^T\|_F^2$ with step size $\eta = 5e - 4$.

The case 2 has much greater $c$ than the case 1, and as the consequence, the loss converges much faster for the case 2, as shown in Fig.2. We see from the right log plot that for the case 1, the bound in Theorem 1 is not a tight characterization of the asymptotic convergence rate, while for the case 2, when $c$ is large, the bound in Theorem 1 is almost tight regarding the asymptotic rate. Clearly for case 1, there are additional factors that contribute to the linear convergence, which would be an interesting research topic in the future work.

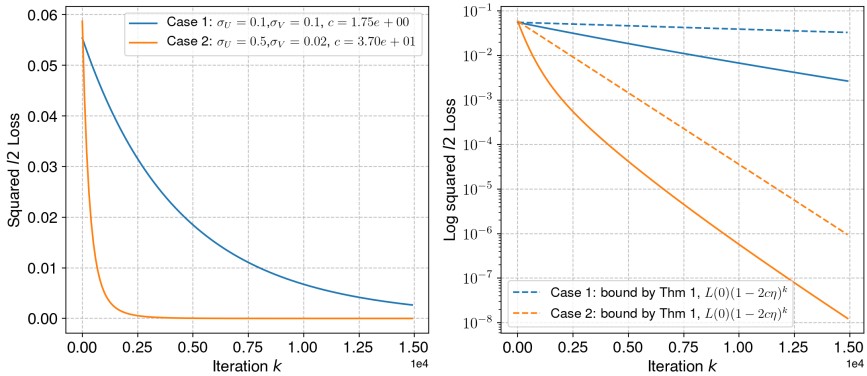

Figure 2: Convergence of gradient descent with different initial level of imbalance, $c :=$ $\sigma_{n+m-1}(U_1^T(0)U_1(0) - V^T(0)V(0))$.

## A.2 IMPLICIT REGULARIZATION ON WIDE SINGLE-HIDDEN-LAYER LINEAR NETWORK

**Generating training data** The synthetic training data is generated as following:

1) For data matrix $X$, first we generate $X \in \mathbb{R}^{n \times D}$ with all the entries sampled from $\mathcal{N}(0, D^{-1})$;

2) For $Y$, we first sample $\Theta \sim \mathcal{N}(0, D^{-1} I_D)$, and $\epsilon \sim \mathcal{N}(0, 0.01^2 I_n)$, then we let $Y = X\Theta + \epsilon$.

**Initialization and Training** We initialize $U(0), V(0)$ with $[U(0)]_{ij} \sim \mathcal{N}(0, h^{-1})$, $[V(0)]_{ij} \sim \mathcal{N}(0, h^{-1})$ and run gradient descent on the averaged loss $\tilde{L} = \frac{1}{n}\|Y - XUV^T\|_F^2$ with step size $\eta = 5e-4$. The training stops when the loss is below $1e-7$. We run the algorithm for various $h$ from 500 to 10000, and we repeat 10 runs for each $h$.

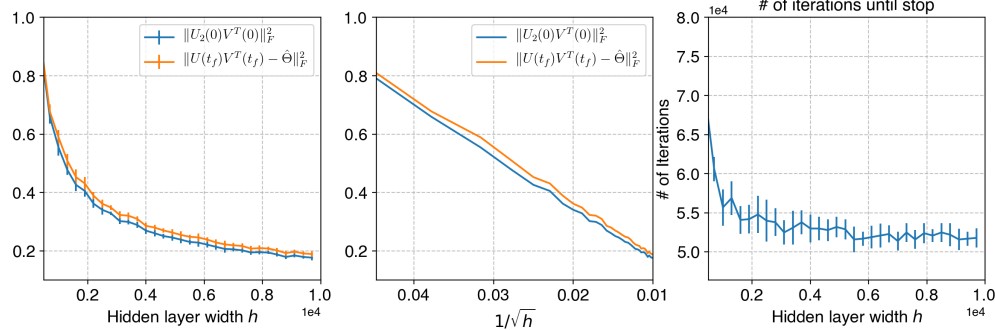

Figure 3: Implicit regularization of wide single-hidden-layer linear network. $\|U_2(0)V^T(0)\|_F^2$ is the initial distance between the end-to-end function to the desired manifold discussed in Section 4.1. $\|U(t_f)V^T(t_f) - \hat{\Theta}\|$ is the distance between the end-to-end function and the min-norm solution when the algorithm stops. The line is plotting the average over 10 runs for each $h$, and the error bar shows the standard deviation.

Clearly, Fig.3 shows that as $h$ increases, the distance between the trained network and the min-norm solution decreases. The middle plot verifies that the distance is indeed $\mathcal{O}(h^{-1/2})$. Lastly, we note that the right plot implies the convergence rate approaches a constant as $h$ increases, which verifies the result in Lemma 1 regarding the imbalance singular value.

# B CONVERGENCE RATE ANALYSIS FOR LINEAR REGRESSION: GENERAL CASE

Suppose the input data matrix $X$ has rank $d \leq \min\{D, n\}$, we write the compact SVD of $X$ as

$$X = W\Sigma_x^{1/2}\Phi_1^T, W \in \mathbb{R}^{n \times d}, \Phi_1 \in \mathbb{R}^{D \times d},$$

and in Section 3 we assume that $d = n < D$. Notice that we always have $W^T W = I$.

Given the compact SVD, we still define $U_1 = \Phi_1^T U$, and write the loss function as

$$
\begin{aligned}
\mathcal{L}(V, U) = \frac{1}{2}\|Y - XUV^T\|_F^2 &= \frac{1}{2}\|Y - W\Sigma_x^{1/2}U_1V^T\|_F^2 \\
&= \frac{1}{2}\|(I_d - WW^T + WW^T)Y - W\Sigma_x^{1/2}U_1V^T\|_F^2 \\
&= \frac{1}{2}\|(I_d - WW^T)Y + W(W^TY - \Sigma_x^{1/2}U_1V^T)\|_F^2 \\
&= \frac{1}{2}\|(I_d - WW^T)Y\|_F^2 + \frac{1}{2}\|W(W^TY - \Sigma_x^{1/2}U_1V^T)\|_F^2 \\
&\qquad + \langle(I_d - WW^T)Y, W(W^TY - \Sigma_x^{1/2}U_1V^T)\rangle_F \\
&= \frac{1}{2}\|(I_d - WW^T)Y\|_F^2 + \frac{1}{2}\|W(W^TY - \Sigma_x^{1/2}U_1V^T)\|_F^2 \\
&= \frac{1}{2}\|(I_d - WW^T)Y\|_F^2 + \frac{1}{2}\|W^TY - \Sigma_x^{1/2}U_1V^T\|_F^2,
\end{aligned}
$$

where the last equality is because that $W^T W = I$, and the second last equality is because that the cross terms equal zero due to $W^T(I_d - WW^T) = 0$.

It is easy to see that $\min_{V,U} \mathcal{L}(V, U) = \frac{1}{2}\|(I_d - WW^T)Y\|_F^2 := \mathcal{L}^*$, which is usually referred as the residue. Similarly to Section.3, we still define the error as $E = W^TY - \Sigma_x^{1/2}U_1V^T$ and one can check that the gradient flow on $\mathcal{L}(V, U)$ yields

$$\dot{V}(t) = E^T(t)\Sigma_x^{1/2}U_1(t), \ \dot{U}_1(t) = \Sigma_x^{1/2}E(t)V(t), \tag{22}$$

in parameter space $(V, U_1)$. Similar to Theorem 1, we have

**Theorem B.1.** *Suppose $h \geq d + m - 1$. Let $V(t), U_1(t), t > 0$ be the trajectory of continuous dynamics* (22) *starting from some $V(0), U_1(0)$. If*

$$\sigma_{d+m-1}\left(U_1^T(0)U_1(0) - V^T(0)V(0)\right) = c > 0,$$

*then for $E(t) = W^TY - \Sigma_x^{1/2}U_1(t)V^T(t)$, we have*

$$\|E(t)\|_F^2 \leq \exp\left(-2\sigma_n(\Sigma_x)ct\right)\|E(0)\|_F^2, \ \forall t > 0.$$

*Additionally, $V(t), U_1(t), t > 0$ converges to some equilibrium point $(V(\infty), U_1(\infty))$ such that $E(V(\infty), U_1(\infty)) = 0$.*

It follows exactly the same proof as for Theorem 1, which is shown in Appendix D, except that the size of $U_1$ and $E$ is now $d \times h$ and $d \times m$ respectively.

To Summarize, for any linear regression problem, Theorem 1 shows that sufficient rank of imbalance guarantees exponential convergence of $\mathcal{L}(t) - \mathcal{L}^*$, where $\mathcal{L}^* = \frac{1}{2}\|(I_d - WW^T)Y\|_F^2$.

# C USEFUL LEMMAS

Before proving Theorem 1 and 2, we state several Lemmas that will be used in the proof.

The first Lemma is the Grönwall's inequality (Grönwall, 1919) in the differential form.

**Lemma C.1** (Grönwall's inequality). *Let $u(t), \beta(t) : [0, +\infty) \rightarrow \mathbb{R}$ be continuous, and $u(t)$ differentiable on $(0, +\infty)$. If*

$$\frac{d}{dt}u(t) \leq \beta(t)u(t), \ \forall t > 0,$$

*then*

$$u(t) \le u(0) \exp\left(\int_0^t \beta(\tau)d\tau\right), \ \forall t > 0.$$

The next Lemma is known as Weyl's Inequality for singular values.

**Lemma C.2** (Weyl's inequality for singular values). *Let $A, B \in \mathbb{R}^{n \times m}$, let $q = \min\{n, m\}$, then*

$$\sigma_{i+j-1}(A + B) \le \sigma_i(A) + \sigma_j(B)$$
$$\sigma_{i+j-1}(AB^T) \le \sigma_i(A)\sigma_j(B^T),$$

*for any $i, j$ satisfying $1 \le i, j \le q$ and $i + j - 1 \le q$.*

The proof can be found in Horn & Johnson (1994, Theorem 3.3.16). Using Weyl's inequality, we state and prove a lemma that is used for proving Theorem 2.

**Lemma C.3.** *Let $A \in \mathbb{R}^{k \times n}, B \in \mathbb{R}^{n \times m}$. Suppose $n \le m$, then*

$$\sigma_i(A)\sigma_n(B) \le \sigma_i(AB),$$

*for $1 \le i \le \min\{k, n\}$.*

*Proof.* We start with the case where $k = n$. When $\sigma_n(B^T) = 0$, the result is trivial. When $\sigma_n(B^T) \ne 0$, we have $BB^\dagger = I$, where $B^\dagger$ is the Moore–Penrose inverse of $B$. By Lemma C.2, it follows that

$$\sigma_i(A) \le \sigma_i(AB)\sigma_1(B^\dagger), \ \forall 1 \le i \le n.$$

Since $\sigma_1(B^\dagger) = \sigma_n^{-1}(B)$, we get the desired inequality.

When $k > n$, we have

$$\sigma_i(A) = \sigma_i\left(\begin{bmatrix} A & 0_{k \times (k-n)} \end{bmatrix}\right) \le \sigma_i(AB)\,\sigma_1(\begin{bmatrix} B^\dagger & 0_{m \times (k-n)} \end{bmatrix}) = \sigma_i(AB)\sigma_1(B^\dagger), \ \forall 1 \le i \le n,$$

which still leads to the desired result.

When $k < n$, consider replacing $A$ with $\begin{bmatrix} A \\ 0_{(n-k) \times n} \end{bmatrix}$, we have

$$\sigma_i(A)\sigma_n(B) = \sigma_i\left(\begin{bmatrix} A \\ 0_{(n-k) \times n} \end{bmatrix}\right)\sigma_n(B) \le \sigma_i\left(\begin{bmatrix} AB \\ 0_{(n-k) \times m} \end{bmatrix}\right) = \sigma_i(AB), \ \forall 1 \le i \le k.$$

$\square$

We also state a trace inequality widely using for solving control problems

**Lemma C.4.** *Suppose for $A, B \in \mathbb{R}^{n \times n}$, $A$ is symmetric and $B$ is positive semidefinite, then*

$$\lambda_n(A)\operatorname{tr}(B) \le \operatorname{tr}(AB) \le \lambda_1(A)\operatorname{tr}(B).$$

*If both $A, B$ are positive semidefinite, then*

$$\sigma_n(A)\operatorname{tr}(B) \le \operatorname{tr}(AB) \le \sigma_1(A)\operatorname{tr}(B).$$

The proof can be found in Sheng-De Wang et al. (1986, Lemma 1).

## D  PROOF OF THEOREM 1

We begin with restating the Theorem.

**Theorem 1** (Convergence of linear networks with sufficient rank of imbalance,restated). *Suppose $h \ge m + n - 1$. Let $V(t), U_1(t), t > 0$ be the trajectory of continuous dynamics (7) starting from some $V(0), U_1(0)$. If*

$$\sigma_{n+m-1}\left(U_1^T(0)U_1(0) - V^T(0)V(0)\right) = c > 0,$$

*then for $E(t)$ defined in (6), we have*

$$\|E(t)\|_F^2 \le \exp\left(-2\sigma_n(\Sigma_x)ct\right)\|E(0)\|_F^2, \ \forall t > 0.$$

*Additionally, $V(t), U_1(t), t > 0$ converges to some equilibrium point $(V(\infty), U_1(\infty))$ such that $E(V(\infty), U_1(\infty)) = 0$.*

*Proof.* For readability we simply write $V(t), U_1(t), E(t)$ as $V, U_1, E$ for most of the proof.

Under (7), the time derivative of error is given by

$$\dot{E} = -\Sigma_x^{1/2} U_1 U_1^T \Sigma_x^{1/2} E - \Sigma_x E V V^T \, .$$

Consider the time derivative of $\|E\|_F^2$,

$$
\begin{aligned}
\frac{d}{dt}\|E\|_F^2 &= \frac{d}{dt}\operatorname{tr}(E^T E) \\
&= -2\operatorname{tr}\left(E^T \Sigma_x^{1/2} U_1 U_1^T \Sigma_x^{1/2} E + E^T \Sigma_x E V V^T\right) \, .
\end{aligned}
\tag{23}
$$

Use the trace inequality in Lemma C.4 to get the lower bound the trace of two matrices respectively as

$$
\begin{aligned}
\operatorname{tr}\left(E^T \Sigma_x^{1/2} U_1 U_1^T \Sigma_x^{1/2} E\right) &= \operatorname{tr}\left(\Sigma_x^{1/2} E E^T \Sigma_x^{1/2} U_1 U_1^T\right) \\
&\geq \sigma_n(U_1 U_1^T)\operatorname{tr}\left(\Sigma_x^{1/2} E E^T \Sigma_x^{1/2}\right) \\
&= \sigma_n(U_1 U_1^T)\operatorname{tr}\left(\Sigma_x E E^T\right) \\
&\geq \sigma_n(U_1 U_1^T)\sigma_n(\Sigma_x)\operatorname{tr}(E E^T) = \sigma_n(U_1 U_1^T)\sigma_n(\Sigma_x)\|E\|_F^2 \, ,
\end{aligned}
\tag{24}
$$

and

$$
\begin{aligned}
\operatorname{tr}\left(E^T \Sigma_x E V V^T\right) &\geq \sigma_m(V V^T)\operatorname{tr}\left(E^T \Sigma_x E\right) \\
&= \sigma_m(V V^T)\operatorname{tr}\left(\Sigma_x E E^T\right) \\
&\geq \sigma_m(V V^T)\sigma_n(\Sigma_x)\operatorname{tr}(E E^T) = \sigma_m(V V^T)\sigma_n(\Sigma_x)\|E\|_F^2 \, .
\end{aligned}
\tag{25}
$$

Combine (23) with (24)(25), we have

$$\frac{d}{dt}\|E\|_F^2 \leq -2\sigma_n(\Sigma_x)\left(\sigma_n(U_1 U_1^T) + \sigma_m(V V^T)\right)\|E\|_F^2 \tag{26}$$

Moreover, we have

$$
\begin{aligned}
&\sigma_n(U_1 U_1^T) + \sigma_m(V V^T) \\
&= \sigma_n(U_1^T U_1) + \sigma_m(V^T V) \\
&= \sigma_n(U_1^T U_1) + \sigma_m(-V^T V) \\
\text{(Lemma C.2)} \quad &\geq \sigma_{n+m-1}(U_1^T U_1 - V^T V) \\
\text{(Imbalance is time-invariant)} \quad &= \sigma_{n+m-1}(U_1^T(0)U_1(0) - V^T(0)V(0)) = c \, ,
\end{aligned}
$$

where the first equality uses the fact that $U_1 U_1^T (V V^T$ resp.) has the same non-zero singular values as $U_1^T U_1 (V^T V$ resp.). Finally we have

$$\frac{d}{dt}\|E\|_F^2 \leq -2\sigma_n(\Sigma_x)c\|E\|_F^2 \, .$$

The result follows by applying Grönwall's inequality, Lemma C.1, which leads to

$$\|E(t)\|_F^2 \leq \exp\left(-2\sigma_n(\Sigma_x)ct\right)\|E(0)\|_F^2, \ \forall t > 0 \, , \tag{27}$$

then the exponential convergence of $E(t)$ is proved.

Regarding the second statement, for the gradient system (7), the parameters $(U_1(t), V(t))$ converge either to an equilibrium point which minimizes the potential $\|E(t)\|_F^2$ or to infinity (Hirsch et al., 1974).

Consider the following dynamics

$$\frac{d}{dt}\begin{bmatrix} V(t) \\ U_1(t) \end{bmatrix} = \underbrace{\begin{bmatrix} 0 & E^T(t)\Sigma_x^{1/2} \\ \Sigma_x^{1/2} E(t) & 0 \end{bmatrix}}_{:=A_Z(t)} \underbrace{\begin{bmatrix} V(t) \\ U_1(t) \end{bmatrix}}_{:=Z(t)}, \tag{28}$$

which is a time-variant linear system. Notice that by Horn & Johnson (2012, Theorem 7.3.3), we have $\|A_Z(t)\|_2 = \|\Sigma_x^{1/2} E(t)\|_2$.

From (28), we have

$$
\begin{aligned}
\frac{d}{dt}\|Z(t)\|_F^2 &= 2 \operatorname{tr}\left(Z^T(t) A_Z(t) Z(t)\right) \\
&= 2 \operatorname{tr}\left(Z(t) Z^T(t) A_Z(t)\right) \\
&\leq 2\|A_Z(t)\|_2 \operatorname{tr}\left(Z(t) Z^T(t)\right) \\
&= 2\|\Sigma_x^{1/2} E(t)\|_2 \|Z(t)\|_F^2 \\
&\leq 2\sigma_1^{1/2}(\Sigma_x)\|E(t)\|_2 \|Z(t)\|_F^2 \leq 2\sigma_1^{1/2}(\Sigma_x)\|E(t)\|_F \|Z(t)\|_F^2 \,.
\end{aligned}
$$

By Grönwall's inequality, Lemma C.1, we have

$$
\|Z(t)\|_F^2 \leq \exp\left(\int_0^t 2\sigma_1^{1/2}(\Sigma_x)\|E(\tau)\|_F d\tau\right) \|Z(0)\|_F^2 \,.
$$

Finally, from (27), we have $\|E(t)\|_F \leq \exp\left(-\sigma_n(\Sigma_x) ct\right)\|E(0)\|_F$, $\forall t > 0$, which leads to

$$
\begin{aligned}
\|Z(t)\|_F^2 &\leq \exp\left(\int_0^t 2\sigma_1^{1/2}(\Sigma_x)\|E(\tau)\|_F d\tau\right) \|Z(0)\|_F^2 \\
&\leq \exp\left(2\sigma_1^{1/2}(\Sigma_x)\|E(0)\|_F \left(\int_0^t \exp\left(-\sigma_n(\Sigma_x) c\tau\right) d\tau\right)\right) \|Z(0)\|_F^2 \\
&\leq \exp\left(2\sigma_1^{1/2}(\Sigma_x)\|E(0)\|_F \left(\int_0^\infty \exp\left(-\sigma_n(\Sigma_x) c\tau\right) d\tau\right)\right) \|Z(0)\|_F^2 \\
&= \exp\left(\frac{2\sigma_1^{1/2}(\Sigma_x)}{c\sigma_n(\Sigma_x)}\|E(0)\|_F\right) \|Z(0)\|_F^2 \,.
\end{aligned}
\tag{29}
$$

Therefore the trajectory $V(t), U_1(t), t > 0$ is bounded, i.e. it can not converge to infinity, then it has to converges to some equilibrium point $(V(\infty), U_1(\infty))$ such that $E(V(\infty), U_1(\infty)) = 0$. $\qquad\square$

## E   COMPARISON WITH THE NTK INITIALIZATION FOR WIDE SINGLE-HIDDEN-LAYER LINEAR NETWORKS

In Section 4.2, we analyzed generalization property of wide single-hidden-layer linear networks under properly scaled random initialization. Our initialization for network weights $U, V$ is different from the typical setting in previous works (Jacot et al., 2018; Du & Hu, 2019; Arora et al., 2019b). In this section, we show that under our setting, the gradient flow is related to the NTK flow by 1) reparametrization and rescaling in time ; 2) proper scaling of the network output. The necessity of output scaling is also shown in Arora et al. (2019b).

In this paper we work with a single-hidden-layer linear network defined as $f : \mathbb{R}^D \to \mathbb{R}^m, f(x; V, U) = VU^T x$, which is parametrized by $U, V$. Then we analyze the gradient flow on the loss function $\mathcal{L}(V, U) = \frac{1}{2}\left\|Y - XUV^T\right\|_F^2$, given the data and output matrix $X, Y$. Lastly, in Section 4.2, we initialize $U(0), V(0)$ such that all the entries are randomly drawn from $\mathcal{N}\left(0, h^{-1}\right)$, where $h$ is the hidden layer width.

Now we define $\tilde{U} := \sqrt{h}U, \tilde{V} := \sqrt{h}V$, then the loss function can be written as

$$
\begin{aligned}
\mathcal{L}(V, U) = \tilde{\mathcal{L}}(\tilde{V}, \tilde{U}) &= \frac{1}{2}\left\|Y - \frac{1}{h}X\tilde{U}\tilde{V}^T\right\|_F^2 = \frac{1}{2}\left\|Y - \frac{\sqrt{m}}{\sqrt{h}}\frac{1}{\sqrt{mh}}X\tilde{U}\tilde{V}^T\right\|_F^2 \\
&= \frac{1}{2}\sum_{i=1}^n \left\|y^{(i)} - \frac{\sqrt{m}}{\sqrt{h}}\frac{1}{\sqrt{mh}}\tilde{V}\tilde{U}^T x^{(i)}\right\|_2^2 \\
&:= \sum_{i=1}^n \left\|y^{(i)} - \frac{\sqrt{m}}{\sqrt{h}}\tilde{f}(x; \tilde{V}, \tilde{U})\right\|_2^2
\end{aligned}
$$

Notice that $\tilde{f}(x; \tilde{V}, \tilde{U}) = \frac{1}{\sqrt{mh}}\tilde{V}\tilde{U}^T x$ is the typical network discussed in previous works (Jacot et al., 2018; Du & Hu, 2019; Arora et al., 2019b). When all the entries of $U(0), V(0)$ are initialized randomly as $\mathcal{N}\left(0, h^{-1}\right)$, the entries of $\tilde{U}(0), \tilde{V}(0)$ are random samples from $\mathcal{N}(0, 1)$, which is the typical choice of initialization for NTK analysis.

However, the difference is that $\tilde{f}(x; \tilde{V}, \tilde{U})$ is scaled by $\frac{\sqrt{m}}{\sqrt{h}}$. In previous work showing non-asymptotic bound between wide neural networks and its infinite width limit (Arora et al., 2019b, Theorem 3.2), the wide neural network is scaled by a small constant $\kappa$ such that the prediction by the trained network is within $\epsilon$-distance to the one by the kernel predictor of its NTK. Moreover, Arora et al. (2019b) suggests $\frac{1}{\kappa}$ should scale as $poly(\frac{1}{\epsilon})$, i.e., to make sure the trained network is arbitrarily close to the kernel predictor, $\kappa$ should be vanishingly small. In our setting, the random initialization implicitly enforces such a vanishing scaling $\frac{\sqrt{m}}{\sqrt{h}}$, as the width of network increases.

Lastly, we show that the gradient flow on $\mathcal{L}(V, U)$ only differs from the flow on $\tilde{\mathcal{L}}(\tilde{V}, \tilde{U})$ by the time scale. Suppose $U, V^1$ follows the gradient flow on $\mathcal{L}(V, U)$ w.r.t. time $t$. Define $\tilde{t} := ht$, we have

$$\frac{d}{d\tilde{t}}\tilde{U} = \sqrt{h}\frac{d}{d\tilde{t}}U = \sqrt{h}\frac{dt}{d\tilde{t}}\frac{d}{dt}U = \frac{1}{\sqrt{h}}\frac{d}{dt}U = -\frac{1}{\sqrt{h}}\frac{\partial}{\partial U}\mathcal{L}(V, U),$$

and similarly we have $\frac{d}{d\tilde{t}}\tilde{V} = -\frac{1}{\sqrt{h}}\frac{\partial}{\partial V}\mathcal{L}(V, U)$.

Now notice that

$$\frac{d}{d\tilde{t}}\tilde{U} = -\frac{1}{\sqrt{h}}\frac{\partial}{\partial U}\mathcal{L}(V, U) = -\frac{1}{\sqrt{h}}X^T(Y - XUV^T)V$$

$$= -\frac{1}{h}X^T\left(Y - \frac{1}{h}X\tilde{U}\tilde{V}^T\right)\tilde{V} = -\frac{\partial}{\partial\tilde{U}}\tilde{\mathcal{L}}(\tilde{V}, \tilde{U}),$$

and

$$\frac{d}{d\tilde{t}}\tilde{V} = -\frac{1}{\sqrt{h}}\frac{\partial}{\partial V}\mathcal{L}(V, U) = -\frac{1}{\sqrt{h}}(Y - XUV^T)^T XU$$

$$= -\frac{1}{h}\left(Y - \frac{1}{h}X\tilde{U}\tilde{V}^T\right)^T X\tilde{U} = -\frac{\partial}{\partial\tilde{V}}\tilde{\mathcal{L}}(\tilde{V}, \tilde{U})$$

Therefore, the gradient flow of $U, V$ on $\mathcal{L}(V, U)$ w.r.t. time $t$ is equivalent to the gradient flow of $\tilde{U}, \tilde{V}$ on $\tilde{\mathcal{L}}(\tilde{V}, \tilde{U})$ w.r.t. a rescaled time $\tilde{t} = ht$.

Another way to see the time scale difference is the following, consider the gradient flow on $\mathcal{L}(V, U)$ w.r.t. time $t$, we have

$$\frac{d}{dt}U(t) = -\frac{\partial}{\partial U}\mathcal{L}(V(t), U(t))$$

$$\Leftrightarrow \frac{1}{\sqrt{h}}\frac{d}{dt}\tilde{U}(t) = -\frac{\partial}{\partial U}\mathcal{L}(V(t), U(t))$$

$$\Leftrightarrow \frac{1}{\sqrt{h}}\frac{d}{dt}\tilde{U}(t) = -\sqrt{h}\frac{\partial}{\partial\tilde{U}}\tilde{\mathcal{L}}(\tilde{V}(t), \tilde{U}(t))$$

$$\Leftrightarrow \frac{d}{dt}\tilde{U}(t) = -h\frac{\partial}{\partial\tilde{U}}\tilde{\mathcal{L}}(\tilde{V}(t), \tilde{U}(t)), \tag{30}$$

where we use the fact that

$$\frac{\partial}{\partial U}\mathcal{L}(V(t), U(t)) = X^T(Y - XU(t)V^T(t))V(t)$$

$$= \frac{1}{\sqrt{h}}X^T\left(Y - \frac{1}{h}X\tilde{U}(t)\tilde{V}^T(t)\right)\tilde{V}(t) = \sqrt{h}\frac{\partial}{\partial\tilde{U}}\tilde{\mathcal{L}}(\tilde{V}(t), \tilde{U}(t)).$$

Similarly we have

$$\frac{d}{dt}V(t) = -\frac{\partial}{\partial V}\mathcal{L}(V(t), U(t)) \Leftrightarrow \frac{d}{dt}\tilde{V}(t) = -h\frac{\partial}{\partial\tilde{V}}\tilde{\mathcal{L}}(\tilde{V}(t), \tilde{U}(t)). \tag{31}$$

---

[1]We write $U(t), V(t)$ as $U, V$ for simplicity. Same for $\tilde{U}(t), \tilde{V}(t)$.

From (30) and (31) we know that the gradient flow on $\mathcal{L}(V, U)$ w.r.t. time $t$ essentially runs the gradient flow on $\tilde{\mathcal{L}}(\tilde{V}, \tilde{U})$ with an accelerated rate by $h$.

Such equivalence through time rescaling suggests that running gradient flow on our setting is $h$ times faster than the NTK one. In Arora et al. (2019b), as we mentioned above, the network is scaled by a small constant $\kappa$ such that the trained network is within $\epsilon$-distance to the kernel predictor by its NTK in terms of the prediction. As a consequence, the convergence rate is scaled by $\kappa^2$, which makes the convergence slower. Therefore, our initialization scheme yields similar result as in Arora et al. (2019b) but the gradient flow is faster. Also, we note that this gap in rate of convergence in not present in (Du & Hu, 2019), which only focuses on providing convergence guarantees of the algorithm. In that case, the network is not scaled by a small $\kappa$, however, the properties of generalization is not studied there.

## F    PROOF OF LEMMA 1 AND THEOREM 2

To prove Lemma 1 and Theorem 2, we use a basic result in random matrix theory

**Lemma F.1.** *Given $m, n \in \mathbb{N}$ with $m \leq n$. Let $A$ be an $n \times m$ random matrix with i.i.d. standard normal entries $A_{ij} \sim \mathcal{N}(0, 1)$. For $\delta > 0$, with probability at least $1 - 2\exp(-\delta^2)$, we have*

$$\sqrt{n} - (\sqrt{m} + \delta) \leq \sigma_m(A) \leq \sigma_1(A) \leq \sqrt{n} + (\sqrt{m} + \delta).$$

The proof can be found in Davidson & Szarek (2001, Theorem 2.13)

In this section, we show more general results under the following random initialization

$$[U(0)]_{ij} \sim \mathcal{N}\left(0, \frac{1}{h^{2\alpha}}\right), \ 1 \leq i \leq D, 1 \leq j \leq h,$$

$$[V(0)]_{ij} \sim \mathcal{N}\left(0, \frac{1}{h^{2\alpha}}\right), \ 1 \leq i \leq m, 1 \leq j \leq h,$$

where $\frac{1}{4} < \alpha \leq \frac{1}{2}$. It is easy to see that $\alpha = \frac{1}{2}$ corresponds to the random initialization scheme shown in Section 4.2, i.e. all the entries of $U(0), V(0)$ are random mean zero Gaussian with $h^{-1}$ variance.

Regarding the imbalance and orthogonality condition, we have the following

**Lemma F.2.** *Let $\frac{1}{4} < \alpha \leq \frac{1}{2}$. Given data matrix $X$. $\forall \delta \in (0, 1)$, $\forall h > h_0 = poly\left(m, D, \frac{1}{\delta}\right)$, with probability at least $1 - \delta$ over random initializations with $[U(0)]_{ij}, [V(0)]_{ij} \sim \mathcal{N}(0, h^{-2\alpha})$, we have all the following hold.*

    *1. (Sufficient rank of imbalance)*

$$\sigma_{n+m}\left(U_1^T(0)U_1(0) - V^T(0)V(0)\right) > h^{1-2\alpha} - 2\frac{\sqrt{m+D} + \frac{1}{2}\log\frac{2}{\delta}}{h^{2\alpha - \frac{1}{2}}},$$

    *2. (Approximate orthogonality condition)*

$$\left\|\begin{bmatrix} V(0)U_2^T(0) \\ U_1(0)U_2^T(0) \end{bmatrix}\right\|_F \leq 2\sqrt{m+n}\frac{\sqrt{m+D} + \frac{1}{2}\log\frac{2}{\delta}}{h^{2\alpha - \frac{1}{2}}},$$

$$\left\|U_1(0)V^T(0)\right\|_F \leq 2\sqrt{m}\frac{\sqrt{m+D} + \frac{1}{2}\log\frac{2}{\delta}}{h^{2\alpha - \frac{1}{2}}}.$$

From the Lemma, we can see why our analysis only applies to the case where $\frac{1}{4} < \alpha \leq \frac{1}{2}$: 1) If $\alpha > \frac{1}{2}$, the lower bound we can obtain for $\sigma_{n+m}\left(U_1^T(0)U_1(0) - V^T(0)V(0)\right)$ will decreases to zero as $h$ increases; 2) If $\alpha \leq \frac{1}{4}$, the orthogonality condition will not be asymptotically satisfied as $h$ increases.

From Lemma F.2, let $\alpha = \frac{1}{2}$, we have

**Lemma 1** (restated). *Given data matrix $X$. $\forall \delta \in (0,1)$, $\forall h > h_0 = poly\left(m, D, \frac{1}{\delta}\right)$, with probability at least $1 - \delta$ over random initializations with $[U(0)]_{ij}, [V(0)]_{ij} \sim \mathcal{N}(0, h^{-1})$, we have all the following hold.*

1. *(Sufficient rank of imbalance)*

$$\sigma_{n+m}\left(U_1^T(0)U_1(0) - V^T(0)V(0)\right) > 1 - 2\frac{\sqrt{m+D} + \frac{1}{2}\log\frac{2}{\delta}}{\sqrt{h}},$$

2. *(Approximate orthogonality condition)*

$$\left\|\begin{bmatrix} V(0)U_2^T(0) \\ U_1(0)U_2^T(0) \end{bmatrix}\right\|_F \leq 2\sqrt{m+n}\frac{\sqrt{m+D} + \frac{1}{2}\log\frac{2}{\delta}}{\sqrt{h}},$$

$$\left\|U_1(0)V^T(0)\right\|_F \leq 2\sqrt{m}\frac{\sqrt{m+D} + \frac{1}{2}\log\frac{2}{\delta}}{\sqrt{h}}.$$

Now we present the proof for Lemma F.2

*Proof of Lemma F.2.* For readability we simply write $U(0), U_1(0), U_2(0), V(0)$ as $U, U_1, U_2, V$.

Consider the matrix $\begin{bmatrix} V^T & U^T \end{bmatrix}$ which is $h \times (m + D)$. Apply Lemma F.1 to matrix $A = h^\alpha \begin{bmatrix} V^T & U^T \end{bmatrix}$, with probability at least $1 - \delta$, we have

$$\sqrt{h} - \left(\sqrt{m+d} + \delta\right) \leq \sigma_{m+d}(h^\alpha \begin{bmatrix} V^T & U^T \end{bmatrix}) \leq \sigma_1(h^\alpha \begin{bmatrix} V^T & U^T \end{bmatrix}) \leq \sqrt{h} + \left(\sqrt{m+d} + \delta\right),$$

which leads to

$$h^{\frac{1}{2}-\alpha} - \frac{\sqrt{m+D} + \frac{1}{2}\log\frac{2}{\delta}}{h^\alpha} \leq \sigma_{m+D}(\begin{bmatrix} V^T & U^T \end{bmatrix}) \leq \sigma_1(\begin{bmatrix} V^T & U^T \end{bmatrix}) \leq h^{\frac{1}{2}-\alpha} + \frac{\sqrt{m+D} + \frac{1}{2}\log\frac{2}{\delta}}{h^\alpha}. \tag{32}$$

Regarding the first inequality, write the imbalance as

$$U_1^T U_1 - V^T V = \begin{bmatrix} V^T & U_1^T \end{bmatrix}\begin{bmatrix} -V \\ U_1 \end{bmatrix} = \begin{bmatrix} V^T & U^T \end{bmatrix}\begin{bmatrix} -I_m & 0 \\ 0 & \Phi_1\Phi_1^T \end{bmatrix}\begin{bmatrix} V \\ U \end{bmatrix}.$$

For $h > \left(\sqrt{m+D} + \frac{1}{2}\log\frac{2}{\delta}\right)^2$, assume event (32) happens, then $\sigma_{m+D}\left(\begin{bmatrix} V^T & U^T \end{bmatrix}\right) \geq h^{\frac{1}{2}-\alpha} - \frac{\sqrt{m+D}+\frac{1}{2}\log\frac{2}{\delta}}{h^\alpha} > 0$, then we have

$$\sigma_{n+m}(U_1^T U_1 - V^T V)$$
$$= \sigma_{n+m}\left(\begin{bmatrix} V^T & U^T \end{bmatrix}\begin{bmatrix} -I_m & 0 \\ 0 & \Phi_1\Phi_1^T \end{bmatrix}\begin{bmatrix} V \\ U \end{bmatrix}\right)$$
$$(\text{Lemma } C.3) \geq \sigma_{n+m}\left(\begin{bmatrix} V^T & U^T \end{bmatrix}\begin{bmatrix} -I_m & 0 \\ 0 & \Phi_1\Phi_1^T \end{bmatrix}\right)\sigma_{m+D}\left(\begin{bmatrix} V \\ U \end{bmatrix}\right)$$
$$= \sigma_{n+m}\left(\begin{bmatrix} -I_m & 0 \\ 0 & \Phi_1\Phi_1^T \end{bmatrix}\begin{bmatrix} V \\ U \end{bmatrix}\right)\sigma_{m+D}\left(\begin{bmatrix} V \\ U \end{bmatrix}\right)$$
$$(\text{Lemma } C.3) \geq \sigma_{n+m}\left(\begin{bmatrix} -I_m & 0 \\ 0 & \Phi_1\Phi_1^T \end{bmatrix}\right)\sigma_{m+D}^2\left(\begin{bmatrix} V \\ U \end{bmatrix}\right)$$
$$= \sigma_{n+m}\left(\begin{bmatrix} -I_m & 0 \\ 0 & \Phi_1\Phi_1^T \end{bmatrix}\right)\sigma_{m+D}^2\left(\begin{bmatrix} V^T & U^T \end{bmatrix}\right)$$
$$= \sigma_{m+D}^2\left(\begin{bmatrix} V^T & U^T \end{bmatrix}\right),$$

where the last equality is due to the fact that $\begin{bmatrix} -I_m & 0 \\ 0 & \Phi_1\Phi_1^T \end{bmatrix}$ has exactly $n + m$ non-zero singular value and all of them are 1.

Therefore when $h > \left(\sqrt{m+D} + \frac{1}{2}\log\frac{2}{\delta}\right)^2$, conditioned on event (32), with probability 1 we have

$$\sigma_{n+m}(U_1^T U_1 - V^T V) \geq \sigma_{m+D}^2\left(\begin{bmatrix} V^T & U^T \end{bmatrix}\right)$$

$$\geq \left( h^{\frac{1}{2}-\alpha} - \frac{\sqrt{m+D} + \frac{1}{2}\log\frac{2}{\delta}}{h^\alpha} \right)^2$$

$$= h^{1-2\alpha} - 2\frac{\sqrt{m+D} + \frac{1}{2}\log\frac{2}{\delta}}{h^{2\alpha-\frac{1}{2}}} + \left( \frac{\sqrt{m+D} + \frac{1}{2}\log\frac{2}{\delta}}{h^\alpha} \right)^2$$

$$> h^{1-2\alpha} - 2\frac{\sqrt{m+D} + \frac{1}{2}\log\frac{2}{\delta}}{h^{2\alpha-\frac{1}{2}}}. \tag{33}$$

Regarding the second and third inequality, using the fact that $\|A\|_F \leq \sqrt{\min\{n,m\}}\|A\|_2, A \in \mathbb{R}^{n\times m}$, we have

$$\frac{1}{\sqrt{m+n}}\left\|\begin{bmatrix} VU_2^T \\ U_1U_2^T \end{bmatrix}\right\|_F \leq \left\|\begin{bmatrix} VU_2^T \\ U_1U_2^T \end{bmatrix}\right\|_2$$

$$= \left\|\begin{bmatrix} I_m & 0 \\ 0 & \Phi_1^T \end{bmatrix}\begin{bmatrix} V \\ U \end{bmatrix}\begin{bmatrix} V^T & U^T \end{bmatrix}\begin{bmatrix} 0 \\ \Phi_2 \end{bmatrix}\right\|_2$$

$$= \left\|\begin{bmatrix} I_m & 0 \\ 0 & \Phi_1^T \end{bmatrix}\left(\begin{bmatrix} V \\ U \end{bmatrix}\begin{bmatrix} V^T & U^T \end{bmatrix} - \eta I_{m+D}\right)\begin{bmatrix} 0 \\ \Phi_2 \end{bmatrix}\right\|_2$$

$$\leq \left\|\begin{bmatrix} V \\ U \end{bmatrix}\begin{bmatrix} V^T & U^T \end{bmatrix} - \eta I_{m+D}\right\|_2, \text{ for any } \eta \in \mathbb{R},$$

where the second equality is by the fact that $\begin{bmatrix} I_m & 0 \\ 0 & \Phi_1^T \end{bmatrix}\begin{bmatrix} 0 \\ \Phi_2 \end{bmatrix} = 0$, and

$$\frac{1}{\sqrt{m}}\left\|U_1V^T\right\|_F \leq \left\|U_1V^T\right\|_2$$

$$= \left\|\begin{bmatrix} 0 & \Phi_1^T \end{bmatrix}\begin{bmatrix} V \\ U \end{bmatrix}\begin{bmatrix} V^T & U^T \end{bmatrix}\begin{bmatrix} I_m \\ 0 \end{bmatrix}\right\|_2$$

$$= \left\|\begin{bmatrix} 0 & \Phi_1^T \end{bmatrix}\left(\begin{bmatrix} V \\ U \end{bmatrix}\begin{bmatrix} V^T & U^T \end{bmatrix} - \eta I_{m+D}\right)\begin{bmatrix} I_m \\ 0 \end{bmatrix}\right\|_2$$

$$\leq \left\|\begin{bmatrix} V \\ U \end{bmatrix}\begin{bmatrix} V^T & U^T \end{bmatrix} - \eta I_{m+D}\right\|_2, \text{ for any } \eta \in \mathbb{R},$$

where the second equality is by the fact that $\begin{bmatrix} 0 & \Phi_1^T \end{bmatrix}\begin{bmatrix} I_m \\ 0 \end{bmatrix} = 0$.

Notice that

$$\left\|\begin{bmatrix} V \\ U \end{bmatrix}\begin{bmatrix} V^T & U^T \end{bmatrix} - \eta I_{m+D}\right\|_2 = \max_i \left|\sigma_i^2(\begin{bmatrix} V^T & U^T \end{bmatrix}) - \eta\right|.$$

Again we let $h > \left(\sqrt{m+D} + \frac{1}{2}\log\frac{2}{\delta}\right)^2$. When event (32) happens, all $\sigma_i^2(\begin{bmatrix} V^T & U^T \end{bmatrix})$ are within the interval $\left[\left(h^{\frac{1}{2}-\alpha} - \frac{\sqrt{m+D}+\frac{1}{2}\log\frac{2}{\delta}}{h^\alpha}\right)^2, \left(h^{\frac{1}{2}-\alpha} - \frac{\sqrt{m+D}+\frac{1}{2}\log\frac{2}{\delta}}{h^\alpha}\right)^2\right]$. Since the choice of $\eta$ is arbitrary, we pick

$$\eta = h^{1-2\alpha} + \left( \frac{\sqrt{m+D} + \frac{1}{2}\log\frac{2}{\delta}}{h^\alpha} \right)^2, \tag{34}$$

which is the mid-point of this interval, then we have

$$\max_i \left|\sigma_i^2(\begin{bmatrix} V^T & U^T \end{bmatrix}) - \eta\right|$$

$$\leq \max\left\{ \left|\left(h^{\frac{1}{2}-\alpha} - \frac{\sqrt{m+D}+\frac{1}{2}\log\frac{2}{\delta}}{h^\alpha}\right)^2 - \eta\right|, \left|\left(h^{\frac{1}{2}-\alpha} + \frac{\sqrt{m+D}+\frac{1}{2}\log\frac{2}{\delta}}{h^\alpha}\right)^2 - \eta\right| \right\}$$

($\eta$ is the mid-point)

$$\leq \left| \left( h^{\frac{1}{2}-\alpha} - \frac{\sqrt{m+D}+\frac{1}{2}\log\frac{2}{\delta}}{h^\alpha} \right)^2 - h^{1-2\alpha} - \left( \frac{\sqrt{m+D}+\frac{1}{2}\log\frac{2}{\delta}}{h^\alpha} \right)^2 \right|$$

$$= 2\frac{\sqrt{m+D}+\frac{1}{2}\log\frac{2}{\delta}}{h^{2\alpha-\frac{1}{2}}}$$

Therefore, when $h > h_0 = \left(\sqrt{m+D}+\frac{1}{2}\log\frac{2}{\delta}\right)^2$, conditioned on event (32), with probability 1, we have

$$\left\| \begin{bmatrix} VU_2^T \\ U_1U_2^T \end{bmatrix} \right\|_F \leq \sqrt{m+n} \left\| \begin{bmatrix} V \\ U \end{bmatrix} \begin{bmatrix} V^T & U^T \end{bmatrix} - \eta I_{m+D} \right\|_2 \leq 2\sqrt{m+n}\frac{\sqrt{m+D}+\frac{1}{2}\log\frac{2}{\delta}}{h^{2\alpha-\frac{1}{2}}},$$

and $\left\| U_1 V^T \right\|_F \leq \sqrt{m} \left\| \begin{bmatrix} V \\ U \end{bmatrix} \begin{bmatrix} V^T & U^T \end{bmatrix} - \eta I_{m+D} \right\|_2 \leq 2\sqrt{m}\frac{\sqrt{m+D}+\frac{1}{2}\log\frac{2}{\delta}}{h^{2\alpha-\frac{1}{2}}},$ (35)

where we choose $\eta$ as in (34). Conditioned on event (32), events (33) and (35) happen with probability 1, hence the probability that both (33) and (35) happen is at least the probability of event (32), which is at least $1-\delta$. □

More generally, for readers' interest, we show that all the non-zero imbalance singular values concentrate to $h^{1-2\alpha}$ as $h$ increases. For the case of $\alpha = \frac{1}{2}$, the singular values concentrate to 1, as suggested by the following

**Claim F.1.** *Let $\frac{1}{4} < \alpha \leq \frac{1}{2}$. Given data matrix $X$. $\forall \delta \in (0,1)$, $\forall h > h_0 = poly\left(m, D, \frac{1}{\delta}\right)$, with probability at least $1 - \delta$ over random initializations with $[U(0)]_{ij}, [V(0)]_{ij} \sim \mathcal{N}(0, h^{-2\alpha})$, we have all the following hold.*

$$\sigma_{n+m}\left(U_1^T(0)U_1(0) - V^T(0)V(0)\right) > \left( h^{\frac{1}{2}-\alpha} - \frac{\sqrt{m+D}+\frac{1}{2}\log\frac{2}{\delta}}{h^\alpha} \right)^2,$$ (36)

$$\sigma_1\left(U_1^T(0)U_1(0) - V^T(0)V(0)\right) \leq \left( h^{\frac{1}{2}-\alpha} + \frac{\sqrt{m+D}+\frac{1}{2}\log\frac{2}{\delta}}{h^\alpha} \right)^2.$$ (37)

*Proof.* For readability we simply write $U(0), U_1(0), V(0)$ as $U, U_1, V$. When the width condition $h > h_0 = poly\left(m, D, \frac{1}{\delta}\right)$ is satisfied. Condition on event (32). The lower bound (36) for the $n+m$-th singular value has been shown by (33).

For the upper bound (37), notice that

$$\sigma_1(U_1^T U_1 - V^T V) = \sigma_1\left( \begin{bmatrix} V^T & U^T \end{bmatrix} \begin{bmatrix} -I_m & 0 \\ 0 & \Phi_1\Phi_1^T \end{bmatrix} \begin{bmatrix} V \\ U \end{bmatrix} \right)$$

$$\leq \sigma_1\left( \begin{bmatrix} -I_m & 0 \\ 0 & \Phi_1\Phi_1^T \end{bmatrix} \right) \sigma_1^2\left( \begin{bmatrix} V^T & U^T \end{bmatrix} \right)$$

$$\leq \sigma_1^2\left( \begin{bmatrix} V^T & U^T \end{bmatrix} \right),$$

where again we use the the fact that $\begin{bmatrix} -I_m & 0 \\ 0 & \Phi_1\Phi_1^T \end{bmatrix}$ has exactly $n+m$ non-zero singular value and all of them are 1.

Condition on event (32), we have

$$\sigma_1\left(U_1^T U_1 - V^T V\right) \leq \sigma_1^2\left( \begin{bmatrix} V^T & U^T \end{bmatrix} \right) \leq \left( h^{\frac{1}{2}-\alpha} + \frac{\sqrt{m+D}+\frac{1}{2}\log\frac{2}{\delta}}{h^\alpha} \right)^2,$$

which is (37). Therefore (36)(37) holds with at least $1-\delta$ probability. □

With Lemma F.2, we have the following result regarding the generalization property of wide single-hidden-layer linear networks. Notice that here the result is presented under random initialization such that all entries of $U(0), V(0)$ are sample from $\mathcal{N}(0, h^{-2\alpha})$, $\frac{1}{4} < \alpha \leq \frac{1}{2}$.

**Theorem F.1.** *Let $\frac{1}{4} < \alpha \leq \frac{1}{2}$. Let $(V(t), U(t), t > 0)$ be a trajectory of continuous dynamics (7). Then, $\exists C > 0$, such that $\forall \delta \in (0,1), \forall h > h_0^{1/(4\alpha-1)}$ with $h_0 = poly\left(m, D, \frac{1}{\delta}, \frac{\sigma_1(\Sigma_x)}{\sigma_n^3(\Sigma_x)}\right)$, with probability $1 - \delta$ over random initializations with $[U(0)]_{ij}, [V(0)]_{ij} \sim \mathcal{N}(0, h^{-2\alpha})$, we have*

$$\|U(\infty)V^T(\infty) - \hat{\Theta}\|_2 \leq 2C^{1/h^{1-2\alpha}}\sqrt{m+n}\frac{\sqrt{m+D} + \frac{1}{2}\log\frac{2}{\delta}}{h^{2\alpha-\frac{1}{2}}}, \tag{38}$$

*where $C$ depends on the data $X, Y$.*

From Theorem F.1, let $\alpha = \frac{1}{2}$, we have

**Theorem 2** (Generalization of wide single-hidden-layer linear network, restated)**.** *Let $(V(t), U(t), t > 0)$ be a trajectory of continuous dynamics (7). Then, $\exists C > 0$, such that $\forall \delta \in (0,1), \forall h > h_0 = poly\left(m, D, \frac{1}{\delta}, \frac{\sigma_1(\Sigma_x)}{\sigma_n^3(\Sigma_x)}\right)$, with probability $1 - \delta$ over random initializations with $[U(0)]_{ij}, [V(0)]_{ij} \sim \mathcal{N}(0, h^{-1})$, we have*

$$\|U(\infty)V^T(\infty) - \hat{\Theta}\|_2 \leq 2C\sqrt{m+n}\frac{\sqrt{m+D} + \frac{1}{2}\log\frac{2}{\delta}}{\sqrt{h}},$$

*where $C$ depends on the data $X, Y$.*

Now we only remain to prove Theorem F.1

*Proof of Theorem F.1.* From the continuous dynamics (7) and Theorem 1, the stationary point $U(\infty), V(\infty)$ satisfy

$$U_1(\infty)V^T(\infty) = \Phi_1^T\hat{\Theta}, \quad U_2(\infty) = U_2(0).$$

Hence we have

$$
\begin{aligned}
\|U(\infty)V^T(\infty) - \hat{\Theta}\|_2 =\ & \|\Phi_1 U_1(\infty)V^T(\infty) + \Phi_2 U_2(\infty)V^T(\infty) - \hat{\Theta}\|_2 \\
=\ & \|\Phi_1\Phi_1^T\hat{\Theta} + \Phi_2 U_2(\infty)V^T(\infty) - \hat{\Theta}\|_2 \\
=\ & \|\Phi_2 U_2(\infty)V^T(\infty)\|_F \\
=\ & \|\Phi_2 U_2(0)V^T(\infty)\|_F = \|U_2(0)V^T(\infty)\|_2 \leq \|U_2(0)V^T(\infty)\|_F.
\end{aligned}
$$

Consider the following dynamics

$$\frac{d}{dt}\begin{bmatrix} V(t)U_2^T(0) \\ U_1(t)U_2^T(0) \end{bmatrix} = \underbrace{\begin{bmatrix} 0 & E^T(t)\Sigma_x^{1/2} \\ \Sigma_x^{1/2}E(t) & 0 \end{bmatrix}}_{:=A_Z(t)}\underbrace{\begin{bmatrix} V(t)U_2^T(0) \\ U_1(t)U_2^T(0) \end{bmatrix}}_{:=Z(t)}, \tag{39}$$

which is a time-variant linear system, and in particular, by Horn & Johnson (2012, Theorem 7.3.3), we have $\|A_Z(t)\|_2 = \|\Sigma_x^{1/2}E(t)\|_2$. Notice that here the $Z(t)$ is different from the one in the proof for Theorem 1.

From (39), we have

$$
\begin{aligned}
\frac{d}{dt}\|Z(t)\|_F^2 =\ & 2\operatorname{tr}\left(Z^T(t)A_Z(t)Z(t)\right) \\
=\ & 2\operatorname{tr}\left(Z(t)Z^T(t)A_Z(t)\right) \\
\leq\ & 2\|A_Z(t)\|_2\operatorname{tr}\left(Z(t)Z^T(t)\right) \\
=\ & 2\|\Sigma_x^{1/2}E(t)\|_2\|Z(t)\|_F^2 \\
\leq\ & 2\sigma_1^{1/2}(\Sigma_x)\|E(t)\|_2\|Z(t)\|_F^2 \leq 2\sigma_1^{1/2}(\Sigma_x)\|E(t)\|_F\|Z(t)\|_F^2.
\end{aligned}
$$

By Grönwall's inequality, Lemma C.1, we have

$$\|Z(t)\|_F^2 \leq \exp\left(\int_0^t 2\sigma_1^{1/2}(\Sigma_x)\|E(\tau)\|_F d\tau\right)\|Z(0)\|_F^2$$

$$\Rightarrow \|Z(t)\|_F \leq \exp\left(\int_0^t \sigma_1^{1/2}(\Sigma_x)\|E(\tau)\|_F d\tau\right)\|Z(0)\|_F \tag{40}$$

Using Lemma F.2, for $h > h_0' := 16\left(\sqrt{m+D} + \frac{1}{2}\ln\frac{2}{\delta}\right)^2$, with probability at least $1 - \delta$ we have all the following.

$$\sigma_{n+m}\left(U_1^T(0)U_1(0) - V^T(0)V(0)\right) > h^{1-2\alpha} - 2\frac{\sqrt{m+D}+\frac{1}{2}\ln\frac{2}{\delta}}{h^{2\alpha-\frac{1}{2}}} \geq \frac{1}{2}h^{1-2\alpha}. \tag{41}$$

$$\|Z(0)\|_F = \left\|\begin{bmatrix} V(0)U_2^T(0) \\ U_1(0)U_2^T(0) \end{bmatrix}\right\|_F \leq 2\sqrt{m+n}\frac{\sqrt{m+D}+\frac{1}{2}\ln\frac{2}{\delta}}{h^{2\alpha-\frac{1}{2}}}, \tag{42}$$

$$\|U_1(0)V^T(0)\| \leq 2\sqrt{m}\frac{\sqrt{m+D}+\frac{1}{2}\ln\frac{2}{\delta}}{h^{2\alpha-\frac{1}{2}}} \tag{43}$$

Here $h_0'$ is larger than one from Lemma 1 because in (41) we want the least non-zero singular value of the imbalance to be further bounded by $\frac{1}{2}$.

From Theorem 1, we have

$$\|E(t)\|_F^2 \leq \exp\left(-2\sigma_n(\Sigma_x)ct\right)\|E(0)\|_F^2,$$

where $c = \sigma_{n+m-1}\left(U_1^T(0)U_1(0) - V^T(0)V(0)\right)$, then by (41), we have

$$\|E(t)\|_F^2 \leq \exp\left(-h^{1-2\alpha}\sigma_n(\Sigma_x)t\right)\|E(0)\|_F^2$$
$$\Rightarrow \|E(t)\|_F \leq \exp\left(-h^{1-2\alpha}\sigma_n(\Sigma_x)t/2\right)\|E(0)\|_F.$$

Finally, from (40), we have

$$\begin{aligned}
\|Z(t)\|_F &\leq \exp\left(\int_0^t \sigma_1^{1/2}(\Sigma_x)\|E(\tau)\|_F d\tau\right)\|Z(0)\|_F \\
&\leq \exp\left(\sigma_1^{1/2}(\Sigma_x)\|E(0)\|_F\left(\int_0^t \exp\left(-h^{1-2\alpha}\sigma_n(\Sigma_x)\tau/2\right)d\tau\right)\right)\|Z(0)\|_F \\
&\leq \exp\left(\sigma_1^{1/2}(\Sigma_x)\|E(0)\|_F\left(\int_0^\infty \exp\left(-h^{1-2\alpha}\sigma_n(\Sigma_x)\tau/2\right)d\tau\right)\right)\|Z(0)\|_F \\
&= \exp\left(2\frac{\sigma_1^{1/2}(\Sigma_x)}{h^{1-2\alpha}\sigma_n(\Sigma_x)}\|E(0)\|_F\right)\|Z(0)\|_F.
\end{aligned} \tag{44}$$

The initial error depends on the initialization but can be upper bounded as

$$\begin{aligned}
\|E(0)\|_F &= \|W^TY - \Sigma_x^{-1/2}U_1(0)V^T(0)\|_F \\
&\leq \|W^TY\|_F + \|\Sigma_x^{-1/2}U_1(0)V^T(0)\|_F \\
&\leq \|Y\|_F + \sigma_n^{-1/2}(\Sigma_x)\|U_1(0)V^T(0)\|_F
\end{aligned}$$

then we can write (44) as

$$\begin{aligned}
\|Z(t)\|_F &\leq \exp\left(2\frac{\sigma_1^{1/2}(\Sigma_x)}{h^{1-2\alpha}\sigma_n(\Sigma_x)}\|Y\|_F\right)\exp\left(2\frac{\sigma_1^{1/2}(\Sigma_x)}{h^{1-2\alpha}\sigma_n^{3/2}(\Sigma_x)}\|U_1(0)V^T(0)\|_F\right)\|Z(0)\|_F \\
&= \left[\exp\left(2\frac{\sigma_1^{1/2}(\Sigma_x)}{\sigma_n(\Sigma_x)}\|Y\|_F\right)\exp\left(2\frac{\sigma_1^{1/2}(\Sigma_x)}{\sigma_n^{3/2}(\Sigma_x)}\|U_1(0)V^T(0)\|_F\right)\right]^{1/h^{1-2\alpha}}\|Z(0)\|_F.
\end{aligned} \tag{45}$$

For the second exponential, we let $h_0 := \max\left\{h_0', 16\frac{\sigma_1(\Sigma_x)}{\sigma_n^3(\Sigma_x)}m\left(\sqrt{m+D}+\frac{1}{2}\ln\frac{2}{\delta}\right)^2\right\}$, then $\forall h > h_0^{1/(4\alpha-1)}$, by (43) we have

$$\exp\left(2\frac{\sigma_1^{1/2}(\Sigma_x)}{\sigma_n^{3/2}(\Sigma_x)}\|U_1(0)V^T(0)\|_F\right) \leq \exp\left(4\frac{\sigma_1^{1/2}(\Sigma_x)}{\sigma_n^{3/2}(\Sigma_x)}\sqrt{m}\frac{\sqrt{m+D}+\frac{1}{2}\ln\frac{2}{\delta}}{h^{2\alpha-\frac{1}{2}}}\right) \leq e. \tag{46}$$

Notice that $h > h_0^{1/(4\alpha-1)}$ also ensures $h > h_0^{1/(4\alpha-1)} \geq h_0 \geq h_0'$, hence the width condition for (41)(42)(43) to hold is satisfied.

Finally by (42)(46), we write (45) as

$$
\begin{aligned}
\|Z(t)\|_F &\leq \left[\exp\left(1 + 2\frac{\sigma_1^{1/2}(\Sigma_x)}{\sigma_n(\Sigma_x)}\|Y\|_F\right)\right]^{1/h^{1-2\alpha}} \|Z(0)\|_F \\
&\leq \underbrace{\left[\exp\left(1 + 2\frac{\sigma_1^{1/2}(\Sigma_x)}{\sigma_n(\Sigma_x)}\|Y\|_F\right)\right]^{1/h^{1-2\alpha}}}_{:=C^{1/h^{1-2\alpha}}} 2\sqrt{m+n}\frac{\sqrt{m+D} + \frac{1}{2}\ln\frac{2}{\delta}}{h^{2\alpha-\frac{1}{2}}} \\
&= 2C^{1/h^{1-2\alpha}}\sqrt{m+n}\frac{\sqrt{m+D} + \frac{1}{2}\ln\frac{2}{\delta}}{h^{2\alpha-\frac{1}{2}}}.
\end{aligned}
$$

Therefore for some $C > 0$ that depends on the data $(X, Y)$, given any $0 < \delta < 1$, when $h > h_0^{1/(4\alpha-1)}$ as defined above, with at least probability $1 - \delta$, we have

$$
\begin{aligned}
\|U(\infty)V^T(\infty) - \hat{\Theta}\|_2 &\leq \|U_2(0)V^T(\infty)\|_F \\
&\leq \sup_{t>0}\|U_2(0)V^T(t)\|_F \\
&\leq \sup_{t>0}\|Z(t)\|_F \leq 2C^{1/h^{1-2\alpha}}\sqrt{m+n}\frac{\sqrt{m+D} + \frac{1}{2}\ln\frac{2}{\delta}}{h^{2\alpha-\frac{1}{2}}}.
\end{aligned}
$$

$\square$

