# OpenReview forum: "On the Explicit Role of Initialization on the Convergence and Generalization Properties of Overparametrized Linear Networks"
_ICLR.cc/2021/Conference — Reject_

### Official Review · AnonReviewer2 · 2020-10-25
**Good result**

**Rating:** 6
**Confidence:** 4

**Review:**

This paper analyzes the convergence of gradient descent optimizing overparametrized linear nn, and proves a exponential convergence rate. Moreover, the paper proposes the distance of the optimizer to the smallest norm solution, which is justified in other papers such as Montanari, etc. as the generalizable solution. Thus the solution that SGD outputs has good generalization as well.
I believe overall the result is good, and the points are stated clearly. I have the following suggestions:

1. If the space is allowed in appendix, the algebra proving that (9) is time invariant can be provided, rather than "one can easily check". This intermediate step is critical for the full proof so I'd like to check it. (with this I can raise score to 6)
2. A sketch of Montanari paper about the property of $\hat \Theta$ can be discussed in the appendix.
3. Regarding the thm, it would be definitely sufficient for the conference if anything can be suggested with RELU activation. In NTK work that's just another kernel so it's easy to extend, but it might be hard here, I'm not sure.
4. More literature review. I think Rong Ge has some papers about the landscape of matrix factorization problem so it's great to compare with them in detail, even if in appendix.
5. In appendix, it's also great to prove that under a certain initialization, what is the expectation/value of high probability of the imbalance singular value.
6. How does amount of data affect generalization bound? I think it's 1/\sqrt{n} in NTK work, any a similar behavior here?

---

> ### Author Response · Authors · 2020-11-25
> **Response to Reviewer 2**
>
> Thank you for your comments on our paper and suggestions. Below are our response to your comments:
>
> 1) **Imbalance matrix and its singular values**:
>
> We greatly appreciate your suggestions and we have added the proof right after we state that the imbalance matrix is time-invariant in Section 3.2.
>
> Regarding the singular values of the imbalance matrix, we have added the result in Appendix F. We show that when the entries are initialized with $\mathcal N(0,h^{-1})$ (setting $\alpha=1/2$ in Claim F.1) all non-zero singular values of the imbalance matrix concentrate to 1 as $h$ increases.
>
> 2) **Property of min-norm solution and generalization bound**:
>
> Thank you for your comment. In our paper, we focus on the implicit regularization of overparametrized networks as the generalization property of our interest. We are interested in how random initialization and overparametrization leads to a regularized solution, close to the minimum norm,  rather than asking how good such a regularized solution is. Making statements on the generalization properties of the minimum norm solution $\hat{\Theta}$ further requires making additional assumptions on the data, that go beyond qualitative conditions such as rank.
> We thus think that a sketch of other papers on the property of min-norm solution $\hat{\Theta}$, which will require an additional set of assumptions, may not be beneficial to the readers.
>
> For the same reason, our analysis on implicit regularization does not directly suggest a generalization bound. But we also agree that it would be interesting to derive generalization bounds for wide networks given additional assumptions on the data.
>
> 3) **Extension to nonlinear networks**:
>
> Indeed, understanding the convergence and generalization properties of networks with nonlinear activation will be of most practical value. However, the case of linear networks has not been fully understood yet. For example, the fact that the imbalance contributes to the exponential convergence has not been shown previously. We believe that we need a deep understanding of simple models, that allow for tighter analyses and counterfactual thinking, in order to thoroughly understand more general cases, with non-linear activations, that are used in deep learning.
>
> Reviewer 1 also asked about the challenges in extending our analysis to ReLU networks. We reiterate next for convenience.
> For the convergence analysis, if the activation is a ReLU, the diagonal terms of the imbalance matrix will be preserved (Du et al. 2018) under differential inclusion. There is still invariance in the imbalance matrix but it is unclear how it contributes to the convergence of the learning dynamics. For the generalization property/implicit regularization, we believe the key challenge would be to identifying the good manifold in terms of generalization, given certain data distribution assumptions.
>
> 4) **Comparison with other literature**:
>
> Thank you for the comment. We carefully read several of Rong's papers on matrix/tensor factorization which we think could be related. But we could not see a direct relationship that merited a detailed comparison. Those papers have settings that are substantially different from ours, for example, they either do not consider a growing overparametrization via width $h$, or they consider different training procedures than gradient flow/descent, thus we don't think a meaningful comparison is possible.

---

### Official Review · AnonReviewer1 · 2020-10-27
**Comments to "On the Explicit Role of Initialization on the Convergence  and Generalization Properties of Overparametrized Linear Networks"**

**Rating:** 9
**Confidence:** 4

**Review:**

#### General Comments
A proper initialization plays an important role in the success of over-parameterized models such as deep neural networks and high dimensional models.  However, the explicit role of initialization in theoretical results of an algorithm has not been stated well to my knowledge.  The main task of this paper is to  present a novel analysis of overparametrized single-hidden layer linear networks, which formally connects initialization, optimization, and overparametrization with generalization performance. Specially,  it is shown that the squared loss converges exponentially at a rate that depends on the level of imbalance of the initialization.
With respect to linear networks, the paper makes the following three main contributions:
（1） The role of initialization of the gradient flow on the convergence is characterized explicitly.
  (2) The stationary point of the gradient flow is sufficiently close to the min-norm solution in the linear case.
（3） Random initialization for large wide  linear networks ensures that the dynamics of the network parameters
      are constrained to a low-dimensional manifold.
Overall, this is a written- well paper with significant novelty.  The results seem interesting in the deep learning theory literature.
#### Specific Comments
(1) For Theorem 2,  the network width is required to be a polynomial of the input dimension D, which may be loose in some practical network structures.  I wonder whether such constrain can be relaxed further? it will be better that some quantitative comparison with those related work is made.
(2) When noisy gradient descent is considered,   is the current analysis  still applicable to the case and similar results can be derived?
(3) If an activation function is added such that the hypothesis class is nonlinear,  is the adopted analysis still valid? if not, what is the additional challenges?

---

> ### Author Response · Authors · 2020-11-25
> **Response to Reviewer 1**
>
> Thank you for your comments on our paper. We are glad you find our results interesting and novel. Below are our responses to your comments:
>
> 1) **Dependence on the input dimension**:
>
> Since we use the basic random matrix theory to prove our concentration result, the input dimension naturally arises as we study the singular values of matrix $\begin{bmatrix}V^T &U^T\end{bmatrix}$. It would be interesting to see whether such dependence is necessary, and we think the answer will be clear once we understand the concentration of the imbalance matrix and other matrices of interest in our analysis, such as $U_1U_2^T,VU_2^T$.
>
> As for the comparison with previous works, we have not seen previously a non-asymptotic bound between the wide linear network to the min-norm solution, to our best knowledge. Therefore there might not be a direct comparison.
>
> 2) **Noisy Gradient Descent**:
>
> Thank you for your question. We believe a similar analysis would work for gradient descent and stochastic gradient descent will sufficiently small step size. While there are additional challenges when moving to discrete-time analysis, we are working in that direction. However, our intuition in this regard suggests that noise will require a faster rate of convergence to counteract/limit the drifting that would move the trajectories away from the "good" manifold.
>
> 3) **Extension to nonlinear networks**:
>
> For the convergence analysis, if the activation is a ReLU, the diagonal terms of the imbalance matrix will be preserved (Du et al. 2018) under differential inclusion. There is still invariance in the imbalance matrix but it is unclear how it contributes to the convergence of the learning dynamics.
>
> For the generalization property/implicit regularization, we believe the key challenge would be to identifying the good manifold in terms of generalization, given certain data distribution assumptions.
>
>
> **References**:
>
> Simon S Du, Wei Hu, and Jason D Lee. "Algorithmic regularization in learning deep homogeneous models: Layers are automatically balanced". In Advances in Neural Information Processing Systems, 2018.

---

### Official Review · AnonReviewer3 · 2020-10-28
**concerns about significance**

**Rating:** 3
**Confidence:** 5

**Review:**

This paper studies the optimization and generalization properties of a two-layer linear network. The considered setting is over-parameterized linear regression where the input dimension is D, number of samples is n<D, and the target dimension is m. The hidden width is h. The paper has two main results. The first result is exponential convergence of gradient flow to global minimum, where the convergence rate depends on the (m+n-1)-th singular value of an "imbalance" matrix. The second result shows that the solution found is close to the minimum L2 norm solution if certain orthogonality assumption is approximately satisfied at initially; then it was shown that if the width h is sufficiently large, then under a random initialization scheme, the solution found is close to the minimum L2 norm solution with a distance $1/\sqrt{h}$.

pros:
The results are not previously known to my knowledge. The proofs appear to be correct as far as I can tell.

cons:
My overall concern is the significance of the results. The results, while correct, do not contribute much to our understandings of optimization and generalization in deep learning. The ways in which the authors interpret the results are unsatisfactory or even misleading.

1) Thm 1 shows a convergence rate of $e^{-ct}$, where $c$ is the (m+n-1)-th singular value of an imbalance matrix. On the appearance this result seems to suggest that a larger $c$ is beneficial for convergence. However I believe this suggestion is incorrect and can be very misleading. Indeed, previous work (e.g. Arora et al. 2018a) has shown linear convergence under zero imbalance ($c=0$), as cited in the paper, but Thm 1 fails to capture that. I think in general this $e^{-ct}$ is a very loose bound that does not capture the real convergence rate (unless the authors can provide convincing evidence that suggests otherwise).

That said, I do think Thm 1 is an interesting theoretical result and the proof is clever. I'm concerned about the practical relevance and the possibly misleading message it sends.

Another weakness is that Thm 1 only considers gradient flow but not gradient descent.

2) Thm 2 and its interpretations are unsatisfactory in a number of ways.

First, we know that just doing a normal linear regression using gradient descent (starting from 0) leads to the minimum L2 norm solution. So now we go through all the trouble in the 2-layer net and finally show we can find a solution that's almost as good as linear regression -- what's the point of doing that?
Of course, one may argue that we are studying a toy model in order to better understand deep learning. However, the main message from this result can be also conveyed in linear regression -- as shown in Sec 4.1, the main step is to find an invariant manifold for gradient flow such that the minimizer in that manifold must be the min-norm solution; for linear regression, such manifold also exists, which is just the span of the data points.

Second, the initialization used ($1/h$ variance in both layers) is unconventional. It's different from the standard 1/fan_in initialization or the NTK parameterization. What happens if we use those more standard initializations? And what happens if we make the initialization smaller, e.g. $1/h^2$, or $1/h^{100}$? Would those change the result? The scale of the initialization is very important in this line of work (such as NTK), so this should be addressed clearly.
(The authors actually claim that as $h\to\infty$ we would get the NTK solution, following Jacot et al (on page 7). I actually don't think Jacot et al.'s work directly implies this, because this paper uses a different initialization scale.)

Third, the authors try to differ this result from all the NTK results, but the theorem is exactly showing that the final solution is close to the NTK solution. Isn't this a bit ironic?

Fourth, the authors claim "this is the first non-asymptotic bound regarding the generalization of linear networks in the global sense." Maybe check out these papers:
Implicit Bias of Gradient Descent on Linear Convolutional Networks,
Implicit Regularization in Matrix Factorization.
Also, many NTK papers also have non-asymptotic bounds. For 2-layer linear networks, one should be able to easily get a bound on the distance of the learned model and the min-norm solution -- might be better than Thm 2.


-------- after rebuttal --------

Thanks to the authors for the response and the updated manuscript. My assessment stays the same, and below are my additional comments.

1. About Appendix E

Thanks for the clarification about the initialization scaling. However, this raises more concern about the significance of the result. In Appendex E, it is shown that the scaling considered in the paper and the NTK scaling lead to the **same** gradient flow dynamics. This suggests that we are actually still in the kernel regime, in contrary to the main motivation and the claims in the paper. (As for the time rescaling issue, it doesn't matter in gradient flow since the difference can be absorbed by rescaling the learning rates.)

Appendix E also mentions that several previous papers used a small multiplier $\kappa$ to make the initial network small. The authors claim that this makes the convergence rate slower. I don't think this affects the convergence rate, but it only affects the width requirement (see e.g. [1]). In fact, in stead of using this multiplier, there is another way to make the output zero without changing the NTK and without requiring a larger width, that is to use an anti-symmetric initialization -- see e.g. [2][3][4][5].

[1] Arora et al. Fine-Grained Analysis of Optimization and Generalization for Overparameterized Two-Layer Neural Networks

[2] Chizat et al. On lazy training in differentiable programming

[3] Hu et al.  Simple and effective regularization methods for training on noisily labeled data with generalization guarantee

[4] Bai and Lee. Beyond linearization: On quadratic and higher-order approximation of wide neural networks

[5] Zhang et al. A type of generalization error induced by initialization in deep neural networks

2. About the motivating questions

This paper proposes to answer two questions in the introduction. The first question is "Is the kernel regime, which requires impractical bounds on the network width, necessary to achieve good generalization?" First, I don't think this paper answers this question since the considered regime is still basically the same as the kernel regime. Second, even if it does, this question itself is not valid, since there are numerous previous theoretical works that study generalization outside the kernel regime, in more interesting settings, e.g. [6][7][8][9] and many more (none of which are mentioned in the paper).

[6] Allen-Zhu and Li. Backward Feature Correction: How Deep Learning Performs Deep Learning

[7] Allen-Zhu and Li. What Can ResNet Learn Efficiently, Going Beyond Kernels?

[8] Wei et al. Regularization Matters: Generalization and Optimization of Neural Nets v.s. their Induced Kernel

[9] Woodworth et al. Kernel and Rich Regimes in Overparametrized Models.

The second main question in the introduction is "Does generalization depends explicitly on acceleration? Or is acceleration required only due to the choosing an initialization outside the good generalization manifold?" I genuinely cannot understand this question.

3. In the updated manuscript the authors state "To the best of our knowledge, this is the first non-asymptotic bound regarding the generalization property of wide linear networks under random initialization in the global sense." This is still false (and insignificant) since the stated result is a direct consequence of previous NTK work.

4. I certainly understand that understanding deep learning is very challenging so it's a natural step to start with simple models. However I think this paper in its current form has limited significance and has major issues in how it discusses previous work, main motivations and contributions, etc., for reasons described in the review.

---

> ### Author Response · Authors · 2020-11-25
> **Response to Reviewer 3 (Part 1)**
>
> Thank you for your comments on our paper. Below are our response to your comments:
>
> 0) **Significance of our results**:
>
> We respectfully disagree with the reviewer's comment about the significance of our work and we have provided a detailed response in the general comments. That being said, we have taken the reviewer's comment very seriously and modified the paper to better articulate the significance of our results. In addition, the reviewer's comments inspired extensions to our analysis that are now also included in appendices E and F. We thank the reviewer for this. Also, we are sorry the reviewer feels the interpretation of our results is misleading. We have thoroughly looked at our paper to identify statements that may have been construed as misleading, and we edited the paper accordingly. We hope the reviewer finds our modified statements concise and accurate.
>
> 1) **Regarding the Convergence Result**:
>
> Both our results and Arora's are sufficient conditions that are valid in complementary regimes. Therefore, one should not expect our results to capture Arora's and vice versa. Specifically, Our result is not showing that $e^{-ct}$ is a tight characterization of the convergence rate of overparametrized linear networks, we never stated this, and we don't think it is true. Rather, we show that the imbalance is a factor that contributes to the exponential convergence of such networks. In the paper, we do not suggest one should artificially make $c$ large for fast convergence, but rather, we show that random initialization together with overparametrization naturally satisfies the rank condition of the imbalance. And this is sufficient for us to further guarantee that the gradient flow stays close to the good generalization manifold.
>
> Previous works (Arora et al. 2018a etc.) have shown very insightful results on the convergence rate when the imbalance is zero. We see our result as a complement to those works because we consider the case where the imbalance is non-zero. Arguably, as most bounds, including those obtained in the related literature, while sufficient, it is not necessary for exponential convergence. We point out, however, that most existing conditions for linear-networks, that are based on requiring an approximately balanced initialization are not satisfied with high probability under random initialization without having the variance of all the entries sufficiently small, which may lead to a poor rate of convergence. We have modified the comments after Theorem 1 to better reflect on our response above.
>
> We also believe similar results can be derived for gradient descent with a sufficiently small step size (In this case the imbalance is not invariant but we should be able to bound its changes), and we are working in that direction.
>
> 2) **Invariant manifold in the overparametrized setting**:
>
> We respectfully disagree with the reviewer's comment questioning the necessity of studying the invariant manifold in the overparametrized setting. As we stated in the global response, fully understanding the simple overparametrized model could shed light on how to analyze more complex models.
> We certainly agree that for the standard linear regression the "good" manifold reduces to the span of the data points. However, unlike the linear regression case,  there is not a clear data-agnostic way to initialize so as to guarantee fast convergence and proximity to the manifold in the overparamterized setting. This is because the zero initialization is in fact a stationary (saddle) point of the gradient flow. As a result, any initialization with small $||U||$ and $||V||$, will be slow to converge, even if it is within the manifold of interest. Our analysis explicitly provides the sufficient condition $V(0)U_2^T(0)=0,U_1(0)U_2^T(0)=0$ to ensure the proximity to the manifold during training. Moreover, we show for wide linear networks with the random initialization, this condition is approximately satisfied for the entire trajectory.

---

> > ### Author Response · Authors · 2020-11-25
> > **Response to Reviewer 3 (Part 2)**
> >
> > 3) **Comparison with NTK and the scale of initialization**:
> >
> > This comment has led to new results and a deeper understanding. We greatly thank the reviewer for this.
> >
> > We believe there is nothing wrong with being unconventional. Our initialization is indeed different from the one used in NTK, but it is arguably better as it achieves the same limiting end-to-end function, but at a faster convergence rate, as our analysis shows. To clarify this issue, we added a detailed comparison of our problem setting with previous works on NTK analysis (See Appendix E.). In particular,  we show that one can relate our model assumptions to the NTK ones by rescaling the parameters and time. This time scaling leads to a slower convergence rate. We also note that our result does not rely on studying the tangent kernel of the network, hence there is a significant difference between our approach and the NTK one.
> >
> > Moreover, we now prove thm 2 in a more general setting where the variance for the entries of $U,V$ is $h^{-2\alpha}$, where $1/4<\alpha\leq 1/2$. The case $\alpha=1/2$, i.e. the variance is $h^{-1}$, is a particular case we consider in the main paper. Finally, we note that our analysis can not make the variance smaller, i.e. $\alpha>1/2$, because the imbalance singular value is vanishingly small as $h$ increases. Please see Appendix F. for the general result for different initialization scale.
> >
> > 4) **Wrong claim in the contribution**:
> >
> > We apologize for our misleading phrase. What we intended to say is "To the best of our knowledge, this is the first non-asymptotic bound regarding the generalization property of wide linear networks under random initialization in the global sense." We have modified the text accordingly. Regarding the non-asymptotic bound from NTK papers, we believe we properly cited related papers (Arora et al., 2019b; Buchanan et al., 2020).
> >
> > **References**:
> >
> > Sanjeev Arora, Nadav Cohen, Noah Golowich, and Wei Hu. "A convergence analysis of gradient descent for deep linear neural networks". In International Conference on Learning Representations, 2018a.
> >
> > Sanjeev Arora, Simon S Du, Wei Hu, Zhiyuan Li, Russ R Salakhutdinov, and Ruosong Wang.
> > "On exact computation with an infinitely wide neural net". In Advances in Neural Information
> > Processing Systems, pp. 8141–8150, 2019b.
> >
> > Sam Buchanan, Dar Gilboa, and John Wright. "Deep networks and the multiple manifold problem".
> > arXiv preprint arXiv:2008.11245, 2020.

---

### Official Review · AnonReviewer4 · 2020-10-28
**Interesting observation and theory, more detailed comparsions and some experiments are needed.**

**Rating:** 5
**Confidence:** 4

**Review:**

This paper proves the convergence rate of gradient flow for training two-layer linear networks. In particular, this paper discusses the connection between initialization, optimization, generalization, and overparameterization. The results show that gradient flow can converge to the global minimum at a rate depending on the level of imbalance of the initialization. Moreover, the authors show that random initialization and overparameterization can implicitly constrain the gradient flow trajectory to converge to a point lying in a low-dimensional manifold, thus guarantees good generalization ability.

This paper is well organized. It is interesting that sufficient imbalance can guarantee global convergence of two-layer linear networks while other papers may require nearly zero initialization or wide enough networks. Besides, my detailed comments are as follows.

One drawback is that this paper still requires that the width be greater than n+m-1 (Theorem 1), while the network width condition proved in some existing works (listed as follows) does not depend on the number of training examples n (although they require random or orthogonal initialization), the authors may need to comment their network width conditions after Theorem 1 (currently the authors only say that “our results is not limited to extremely wide networks with random initialization’’).

[1] Du, S. S., & Hu, W., Width provably matters in optimization for deep linear neural networks. arXiv preprint arXiv:1901.08572.

[2] Hu, W., Xiao, L., & Pennington, J., Provable Benefit of Orthogonal Initialization in Optimizing Deep Linear Networks. In International Conference on Learning Representations.

[3] Zou, D., Long, P. M., & Gu, Q., On the Global Convergence of Training Deep Linear ResNets. In International Conference on Learning Representations.

The authors prove that the limit of gradient flow can be sufficiently close to the minimum-norm solution if the neural network is sufficiently wide. This conclusion is good and of certain importance to understand the optimization path of training linear networks. However, if the data matrix X is of full rank and D<n, the training objective is strongly convex. In this case, there is only one minimum, thus the convergence result Theorem 1 can directly imply the parameter convergence results in Theorem 2.

Some experiments may be needed to verify the theory. In particular, theorem 1 only provides an upper bound result, thus cannot fully characterize the effect of the imbalance on the convergence. The authors may try initializations with different imbalances and plot the convergence curves to demonstrate the results in Theorem 1. Additionally, results in Theorem 2 may also need to be verified in experiments.

So far I can only see that the imbalance plays an important role in training two-layer linear networks, can you extend this to multi-layer cases? Will the imbalance at the initialization together with sufficient overparameterization still guarantee the convergence of gradient flow?

---

> ### Author Response · Authors · 2020-11-25
> **Response to Reviewer 4**
>
> Thank you for your comments on our paper. We are glad that you find our result interesting. Below are our response to your comments:
>
> 1) **On the width requirement $h>n+m-1$**:
>
> The reviewer raises an interesting point that we had discussed in the appendix, where we show that the width requirement can be relaxed depending on the rank of the data matrix X, which consistent with previous literature. Specifically, in Section 2 we show that when the data matrix $X$ has full rank and $n<D$, the width requirement is $h>n+m-1$. However, in appendix B we show that for rank deficient data matrix $X$ with $rank(X)=d\leq n<D$, the width requirement becomes $h>d+m-1$. We apologize for not presenting the result in full generality in the main paper. For the works listed by the reviewer, the width requirement scales up as the rank of data matrix $X$ increases, but we note that there is hardly a direct comparison because of the different settings. We have refined the remarks after Theorem 1 to address your concern.
>
> 2) **Over-determined linear regression case**
>
> Correct. When the data matrix $X$ satisfies $n>D$, the regression problem is over-determined. Theorem 1 implies exponential convergence to a global minimum of the loss function, whose end-to-end function corresponds to the unique solution of the regression problem. However, this is not the regime we are interested in since, as the reviewer points out, in that case, the analysis is straightforward.
>
> 3) **Numerical Verification and Conservativeness of our Imbalance Bound**:
>
> Thank you for your suggestion. We added numerical validations for thm 1 and thm 2 in Appendix A. Indeed, the bound on the convergence rate based on the Imbalance is only an upper bound. The convergence rate does depend on several aspects of the initialization as the existing literature points out. Our imbalance bound is neither better nor worse than the previous bound. Rather it complements existing results by showing a different regime that ensures exponential convergence. Interestingly, this regime is in fact more aligned with standard practices in Deep Learning, as the alternative analysis requires approximate balancedness, which is not satisfied with high probability under random initialization.
>
> 4) **Extension to multi-linear cases**:
>
> We believe imbalance also plays a role in training multi-layer networks, but rigorously showing so is the subject of future research. In particular, the notion of imbalance for multi-layer networks does exist, but it is unclear under which conditions on the imbalance the exponential convergence is guaranteed. Moreover, similar to the single-hidden-layer case, we would like to find conditions that are naturally satisfied by random initialization along with overparametrization. Even if we had an answer to all these questions, we do not think we could concisely fit them with the existing results.

---

### Decision · Program_Chairs · 2021-01-07
**Final Decision**

**Decision:**

Reject

**Comment:**

The authors provide a new analysis of learning of two-layer linear networks with gradient flow, leading to some novel optimization and generalization guarantees incorporating a notion of the imbalance in the weights.  While there was some diversity of opinion, the prevailing view was that the results were not sufficiently significant for publication in ICLR.